# Learning the Electronic Hamiltonian of Large Atomic Structures

Chen Hao Xia [* 1]   Manasa Kaniselvan [* 1]   Alexandros Nikolaos Ziogas [1]   Marko Mladenović [1]   Rayen Mahjoub [1]
Alexander Maeder [1]   Mathieu Luisier [1]

## Abstract

Graph neural networks (GNNs) have shown promise in learning the ground-state electronic properties of materials, subverting *ab initio* density functional theory (DFT) calculations when the underlying lattices can be represented as small and/or repeatable unit cells (i.e., molecules and periodic crystals). Realistic systems are, however, non-ideal and generally characterized by higher structural complexity. As such, they require large (10+ Å) unit cells and thousands of atoms to be accurately described. At these scales, DFT becomes computationally prohibitive, making GNNs especially attractive. In this work, we present a strictly local equivariant GNN capable of learning the electronic Hamiltonian ($\boldsymbol{H}$) of realistically extended materials. It incorporates an *augmented partitioning* approach that enables training on arbitrarily large structures while preserving local atomic environments beyond boundaries. We demonstrate its capabilities by predicting the electronic Hamiltonian of various systems with up to 3,000 nodes (atoms), 500,000+ edges, ~28 million orbital interactions (nonzero entries of $\boldsymbol{H}$), and ≤0.53% error in the eigenvalue spectra. Our work expands the applicability of current electronic property prediction methods to some of the most challenging cases encountered in computational materials science, namely systems with disorder, interfaces, and defects.

## 1. Introduction

Graph neural networks (GNNs) have proven capable of learning properties that are functions of molecular and atomic structures, and thus easily represented by graph-structured data (Veličković et al., 2018). They have already enabled new directions of research previously thought computationally unaffordable. More recently, GNNs have been applied to predict electronic-level properties. The most general of such properties is the ground-state Hamiltonian matrix $\boldsymbol{H}$, which, if expressed in a localized basis, can be decomposed into sub-matrices $\boldsymbol{H}_{i,j}$ that encode the coupling between the sets of basis elements located on atoms $i$ and $j$. These coupling terms depend on the nature (atomic type) and relative coordinates of the local environment.

Previous works on electronic property prediction focused on the cases of molecules (Zhong et al., 2023; Yu et al., 2023b; Bai et al., 2021) and ordered materials (Li et al., 2022; Gong et al., 2023; Wang et al., 2024a) whose graph representations are fairly small; typical molecules contain only a few atoms and all relevant structural information in crystalline materials can be captured by translating a small unit cell. The electronic properties of such systems can be computed "exactly" with density functional theory (DFT) within a few minutes on a single CPU.

Real materials, however, are not made with the repetition of small unit cells. The unavoidable presence of defects, for example, doping atoms, vacancies, strain, compositional variations, or lattice mismatch (Ducry et al., 2020; Kaniselvan et al., 2023; Strand et al., 2018), requires large unit cells composed of up to a few thousand atoms to capture the associated structural disorder (Repa & Fredin, 2023). Accurate DFT simulations of such non-ideal materials are prohibitively expensive, even on today's largest supercomputers. The prospect of applying machine learning solutions to handle such systems is thus particularly attractive and of high relevance in computational materials science.

Here, we adapt equivariant GNN approaches to learn the ground-state Hamiltonian $\boldsymbol{H}$ of this more realistic class of atomic structures. As main contributions:

- We develop a strictly local GNN-based architecture tailored for electronic property prediction from small to large scales. Our network leverages efficient SO(2)-convolutions and multi-headed attention, with learnable embeddings for node and edges that are mapped to diagonal ($\boldsymbol{H}_{i,i}$) and off-diagonal ($\boldsymbol{H}_{i,j}$) blocks.

---

[*]Equal contribution  [1]Integrated Systems Laboratory, Department of Information Technology and Electrical Engineering, ETH Zurich, Zurich, Switzerland. Correspondence to: Chen Hao Xia <chexia@iis.ee.ethz.ch>, Manasa Kaniselvan <mkaniselvan@iis.ee.ethz.ch>.

*Proceedings of the 42$^{nd}$ International Conference on Machine Learning*, Vancouver, Canada. PMLR 267, 2025. Copyright 2025 by the author(s).

- We propose an efficient augmented partitioning method that incorporates virtual nodes/edges to enable partitioning of input graphs while maintaining their exact connectivity. Arbitrarily large graphs can then be decomposed into independent partitions that fit into GPU memory during training without compromising the achievable testing accuracy.

We combine our network and *augmented partitioning* approach to treat a custom dataset of three large atomic structures in their amorphous phase. As amorphous materials encompass all aforementioned defect types, they are ideally suited to test and validate our model. In particular, we achieve 2.17-2.58 meV prediction accuracies on unseen samples, matching the ranges of previous studies for materials with orders of magnitude fewer atoms (Wang et al., 2024b). We also demonstrate how this error translates into downstream applications by matching the eigenvalues of $\boldsymbol{H}$ to within 0.53% relative L1 error on structures with 3,000 atoms, which require several (3.65) hours to compute using DFT. Our work advances applications of equivariant GNNs towards practical use cases in computational materials science.

## 2. Background & related work

The electronic properties of a material refer to its set of energy levels ($\varepsilon$) and wavefunctions ($\psi$) that electrons can occupy. They correspond to the eigenvalues and eigenvectors of the Hamiltonian matrix $\boldsymbol{H}$ describing the atomic system of interest. This quantity is a function of the location (relative positions $\{\boldsymbol{r}_i\}$) and identity (atomic numbers $\{Z_i\}$) of all constituent atoms $\{i\}$ ((Hohenberg & Kohn, 1964)). Therefore, predicting the electronic properties of materials consists of learning the mapping $F : \{\boldsymbol{r}_i, Z_i\} \rightarrow \boldsymbol{H}$ between the atomic structure and the elements of the corresponding Hamiltonian matrix, before diagonalizing $\boldsymbol{H}$ to obtain $\varepsilon$ and $\psi$ (**Fig. 1**).

The entries of the ground-state Hamiltonian matrix $\boldsymbol{H}$ are typically computed from first principles with DFT ((Kohn & Sham, 1965)). In several widely used codes, the wavefunctions are expanded into a basis $|\varphi\rangle$ of non-orthogonal atomic orbitals localized around atomic positions, each built, for example, from contracted Gaussian functions ((Kühne et al., 2020; Neese, 2011)). These orbitals transform like spherical harmonics under rotation $\hat{\boldsymbol{r}} \rightarrow \hat{\boldsymbol{r}}'$: $Y^l_m(\hat{\boldsymbol{r}}') = \sum_{m'} \boldsymbol{D}^l_{mm'}(\boldsymbol{R})Y^l_{m'}(\hat{\boldsymbol{r}})$. Here, $Y^l_m$ is the spherical harmonic of degree $l$ and order $m \in \{-l, \ldots, l\}$. $\boldsymbol{D}^l_{mm'}(\boldsymbol{R})$ is the Wigner-D matrix of degree $l$ corresponding to the rotation $\boldsymbol{R}$, which transforms the spherical harmonic $\hat{\boldsymbol{r}}$ and $\hat{\boldsymbol{r}}'$ are normalized direction vectors.

The localized nature of the basis states leads to finite spatial overlaps between them. The resulting Schrödinger-like

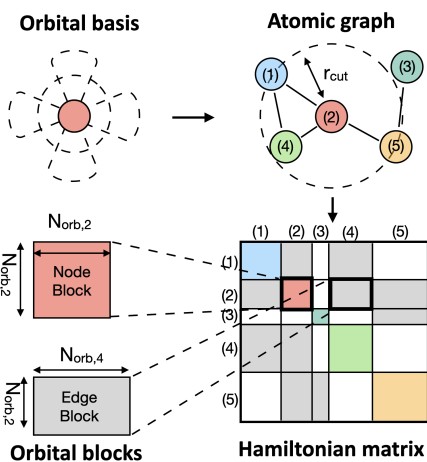

*Figure 1.* Schematic of the mapping between the atomic graph and the blocks of the Hamiltonian matrix $\boldsymbol{H}$ in the localized orbital basis of choice. Each orbital block represents the couplings between atomic orbitals on the same atom ($\boldsymbol{H}_{i,i}$, diagonal) or between two different atoms within r$_{cut}$ ($\boldsymbol{H}_{i,j}$, off-diagonal).

equation at the core of DFT takes the form of a generalized eigenvalue problem: $\boldsymbol{H}\psi = \varepsilon\boldsymbol{S}\psi$. Here, the Hamiltonian matrix $\boldsymbol{H}^{(N \times N)}$ has entries $\boldsymbol{H}_{i,j} = \langle\varphi_i|\hat{H}(\boldsymbol{r})|\varphi_j\rangle$ where $\hat{H}(\boldsymbol{r})$ is the so-called Hamiltonian operator. The Overlap matrix $\boldsymbol{S}^{(N \times N)}$ is defined as $\boldsymbol{S}_{i,j} = \langle\varphi_i|\varphi_j\rangle$. They are both coarse-grained matrices of size $N = \sum_k N^k_{atoms} \cdot N^k_{orb}$, where $N_{atoms}$ is the number of atoms, $N_{orb}$ the number of orbitals per atom, and $k$ indices over the different atomic species found in the system. Note that $\boldsymbol{S}$ reduces to the identity matrix in case of an orthogonal basis $|\varphi\rangle$. Otherwise, it can be directly computed from the basis as the problem's physics does not influence it.

The Hamiltonian matrix can be decomposed into sub-matrices $\boldsymbol{H}_{i,j}$ of size $(N^i_{orb} \times N^j_{orb})$, each describing the interactions between all basis elements (orbitals) on atoms $i$ and $j$. Diagonal blocks ($\boldsymbol{H}_{i,i}$) are the interactions between orbitals on the same atom. When represented on a local basis, the matrix is near-sighted; the interactions between orbitals on different atoms decay exponentially with increasing interatomic distance. Since an atomic orbital basis is used, the sub-matrices are also equivariant under rotation of the atomic bonds, with their transformation properties related by the Wigner-D matrix.

### 2.1. Challenges unique to disordered materials

Computing the electronic properties of disordered materials with DFT still requires defining a repeatable 'unit cell' that is translated through space to construct the desired atomic system. Periodic boundaries are applied to the surface of

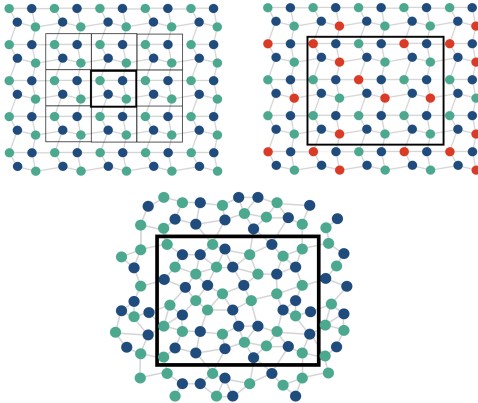

*Figure 2.* Illustration of the differences between ideal periodic (left) and compositionally (middle) or structurally (right) disordered materials, whose features can only be described by large unit cells. In all examples, a black box delimits the smallest repeatable unit cell, while the circles/lines correspond to atoms/bonds.

this cell to avoid dangling bonds. However, the assumed periodicity can alter the disordered nature of the material if the repeating unit cell is too small. If atoms can interact with all their periodic images, non-physical phenomena may arise such as the formation of coherent electronic states across cells, drastically affecting the material properties. Small unit cells thus cannot accurately represent the physics of disordered systems. To avoid such unrealistic scenarios and better approximate disorder, 'large-enough' unit cells should be constructed, with dimensions ranging from 10 Å (Repa & Fredin, 2023) to a few nanometers (Ducry et al., 2020). Generating the Hamiltonian matrix $H$ of these systems with DFT involves tens to hundreds of self-consistent field (SCF) loops, each requiring the diagonalization of an intermediate $H$, the solution of Poisson's equation, and the creation of a new Hamiltonian matrix. As the first operation scales with $\mathcal{O}(N_{atoms}^3)$, analyzing electronic properties for large disordered systems (or different representations of disorder for the same material) is often computationally unaffordable.

## 2.2. Hamiltonian prediction models

Only a few studies have attempted to directly predict the Hamiltonian matrix $H$ of a given material with a GNN rather than fitting its invariant quantities such as the total energy. The key is to constrain the solution space by leveraging prior knowledge of physical symmetries, e.g., rotational equivariance of orbital blocks.

In such equivariant GNNs, the predicted Hamiltonian rotates along with the input (Yu et al., 2023b; Zhang et al., 2024; Batatia et al., 2023; Gong et al., 2023), which requires maintaining SO(3)-equivariance within the model.

In other words, all network operations $f$ acting on input embedding $x^l$ of degree $l$ must satisfy: $f(\boldsymbol{D}^l(\boldsymbol{R}) \cdot \boldsymbol{x}^l) = \boldsymbol{D}^l(\boldsymbol{R}) \cdot f(\boldsymbol{x}^l)$. The resulting networks are trained using Message Passing (MP), where each MP layer works as follows: An atom $i$ receives input messages from all its neighboring source atoms $j$. Each input message goes through convolution operations that combine features with different $l$ while preserving equivariance; a specific output embedding $\boldsymbol{x}_{ji}^{l_3}$ of degree $l_3$ can be computed through: $\boldsymbol{x}_{ji}^{l_3} = \sum_{l_1,l_2} \boldsymbol{x}_j^{l_1} \otimes_{l_1,l_2}^{l_3} h_{l_1,l_2,l_3} Y^{l_2}(\hat{\boldsymbol{r}}_{ji})$. Here, $\hat{\boldsymbol{r}}_{ji}$ is a normalized vector indicating the direction of the edge connecting the atoms $j$ and $i$, and $h$ is a set of trainable weights. The sum runs over tensor products which take $\boldsymbol{x}_j$ (a source input embedding of degree $l_1$) and $Y^{l_2}$ (a filter spherical harmonic embedding of degree $l_2$) and produce the output embedding:

$$(\boldsymbol{x}_j^{l_1} \otimes_{l_1,l_2}^{l_3} Y^{l_2}(\hat{\boldsymbol{r}}_{ji}))_{m_3}^{l_3} = \sum_{m_1,m_2} C(\boldsymbol{x}_j^{l_1})_{m_1} h_{l_1,l_2,l_3} Y_{m_2}^{l_2}(\hat{\boldsymbol{r}}_{ji}),$$

where $C = C_{(l_1,m_1),(l_2,m_2)}^{l_3,m_3}$ are the Clebsch-Gordan coefficients that are indexed by the order $m$ and degree $l$ of the input, filter, and output embeddings. The combination of feature ($x$) and geometric ($\hat{r}$) information along each edge encodes both the identity and structure of the system. These 'Tensor Field Networks' (TFNs) (Thomas et al., 2018) achieve state-of-the-art accuracy on small molecule (Yu et al., 2023b) and crystalline (Gong et al., 2023) datasets. However, they are also computationally expensive. The network training scales with $\mathcal{O}(l_{max}^6)$, where $l_{max}$ is the maximum degree of the angular momentum considered. As a consequence, E(3)-equivariant tensor product networks are difficult to apply beyond a few atoms (Zhang et al., 2024).

Recently, the cost of training equivariant GNNs has been significantly reduced by combining the benefits of data rotation and equivariant network operations. These approaches take advantage of the fact that when edges are rotated to align with a fixed axis ($y$ or $z$, depending on convention), the only non-zero spherical harmonic components are those of order $m = 0$. By keeping track of the bond vectors and performing internal spherical rotations, complex SO(3) convolutions can thus be reduced to SO(2) linear convolution operations (Passaro & Zitnick, 2023). Furthermore, under these conditions, the Clebsch-Gordan coefficients exhibit a predictable sparsity pattern (non-zero only when $m_3 = \pm m_1$). Altogether, the scaling reduces to $\mathcal{O}(l_{max}^3)$ (Wang et al., 2024a), speeding up training, while allowing for higher-order angular momenta ($l_{max}$) and more parameters to capture finer, more complex details of the surrounding environment. An advanced SO(2) convolution network was developed by (Passaro & Zitnick, 2023) (eSCN), and was further expanded by (Liao et al., 2023) (EquiformerV2)

with the inclusion of equivariant attention. An implementation of this architecture on Hamiltonians by (Wang et al., 2024a) achieved better performance on custom crystalline 2D-material datasets compared to previous tensor field and invariant networks. However, the prediction of large disordered structures still remains an open problem in literature.

## 3. Methods

We first present an equivariant architecture that is custom-made for large-scale Hamiltonian prediction. A high-level overview is shown in **Fig. 3(a)-(b)**. Relevant implementation details and ablation studies are presented in **Section 4** and **Appendix B**.

### 3.1. Orbital block processing

Each block $\boldsymbol{H}^{ij}_{\alpha\beta}$ of the Hamiltonian matrix represents the interaction (coupling) between the spherical harmonics of degree $l_\alpha$ on atom $i$ and that of degree $l_\beta$ on atom $j$. Mathematically, it can be cast into the tensor product $l_\alpha \otimes l_\beta$ of length $(2l_\alpha + 1) \times (2l_\beta + 1)$ between the uncoupled angular momentum eigenstates $l_\alpha$ and $l_\beta$. Each of these tensor products can be decomposed into a direct sum ($\oplus$) of coupled angular momentum eigenstates $|L, M\rangle$, where $L$ ranges from $|l_\alpha - l_\beta|$ to $|l_\alpha + l_\beta|$: $T(l_\alpha \otimes l_\beta) = |l_\alpha - l_\beta| \oplus ... \oplus (l_\alpha + l_\beta)$. The transformation $T$ from the uncoupled to the coupled basis is performed using a matrix of Clebsch-Gordon coefficients. For example, the coefficient for a specific $|L, M\rangle$ coupled angular momentum eigenstate component is given by:

$$|L, M\rangle = \sum_{m_\alpha=-l_\alpha}^{+l_\alpha} \sum_{m_\beta=-l_\beta}^{+l_\beta} C^{(L,M)}_{(l_\alpha,m_\alpha)(l_\beta,m_\beta)} |l_\alpha, m_\alpha\rangle |l_\beta, m_\beta\rangle,$$

Applying a rotation $\mathbf{R}$ to the resulting $\boldsymbol{H}^{ij}_{\gamma\delta}$ then consists of independently applying the corresponding Wigner-D transformation $W^l_D(\mathbf{R})$ to each of its subspaces of degree $l$. To minimize the number of transformations, all orbital blocks are initially transformed into the coupled basis as a pre-processing step (**Fig. 3(a)**).

### 3.2. Network architecture

The graph's nodes/edges are first initialized with embeddings of shape $(N_n, (l_{max} + 1)^2, E_n)/(N_e, (l_{max} + 1)^2, E_e)$, where $N_{n/e}$ is the number of nodes/edges, $E_{n/e}$ the dimension of the node/edge embeddings, and $l_{max}$ the maximum degree of the features. The $l = 0$ channels of the node embeddings are initialized with atomic numbers, while those of the edge embeddings are initialized with the scalar distance between the two connecting nodes, expanded in the chosen basis, here, contracted Gaussian functions. All other

components are initially set to 0.

During the node update phase, each node $i$ receives messages from all its neighbors $j$, consisting of the concatenated embeddings $\boldsymbol{n}_i$, $\boldsymbol{n}_j$, and $\boldsymbol{e}_{ji}$. To enable fast tensor product operations in large atomic structures, we adopt eSCN convolutions (Passaro & Zitnick, 2023), with incoming messages rotated to align with the $z$-axis during these operations. Non-linearity is then introduced through a gate activation layer. On top of this, we also include multi-headed attention mechanisms, as in EquiformerV2 (Liao et al., 2023), which allows our network to learn better from highly dense local atomic environments with varying node degree. Finally, the resulting output messages are rotated back to their original orientations, aggregated onto the node $i$, and passed through a feedforward network to update its output embedding $\boldsymbol{n}_i$ corresponding to onsite (diagonal) Hamiltonian blocks.

To learn offsite (non-diagonal) blocks, we use concepts from the Hamiltonian prediction networks in (Gong et al., 2023),(Wang et al., 2024a), and introduce learnable embeddings for every edge, defined between pairs of atoms located within a distance $r_{cut}$ from each other. The updated node embeddings are used to update the edges through a similar process, without the attention layer. The predicted outputs are then post-processed back into the uncoupled basis with the reverse transformation: $T^{-1}(\boldsymbol{H}^{ij}_{\gamma\delta}) = \boldsymbol{H}^{ij}_{\alpha\beta}$, such that they can be used to reconstruct the Hamiltonian matrix block-by-block.

### 3.3. Training on augmented partitions

A notable difference between our application and that of standard GNNs in computational materials resides in the density of interatomic interactions. Many types of physical interactions are short-range and can be captured with a limited receptive field (Batatia et al., 2025). However, despite the near-sightedness property of Gaussian functions, orbitals located on different atoms can interact with each other over distances exceeding $\sim 10$ Å (see **Appendix F**), giving rise to specific nonzero off-diagonal blocks in the Hamiltonian matrix. They must be taken into account by increasing $r_{cut}$. The graph representation of these structures is thus densely connected, with $> 10^2$ edges per node.

The combination of large graphs, dense connectivity, and large tensor representations of nodes and edges results in high memory consumption and long computation times per epoch during training (see **Appendix H.2**). If a distributed computing environment was used, the communication of intermediate values during the forward pass would incur significant overhead, impeding scalability (Wan et al., 2022). At the same time, graphs cannot be arbitrarily partitioned by removing inter-atomic connections. Doing so misinforms the network as it tries to fit the target data while aggregating inputs through an incorrect/incomplete graph structure. The

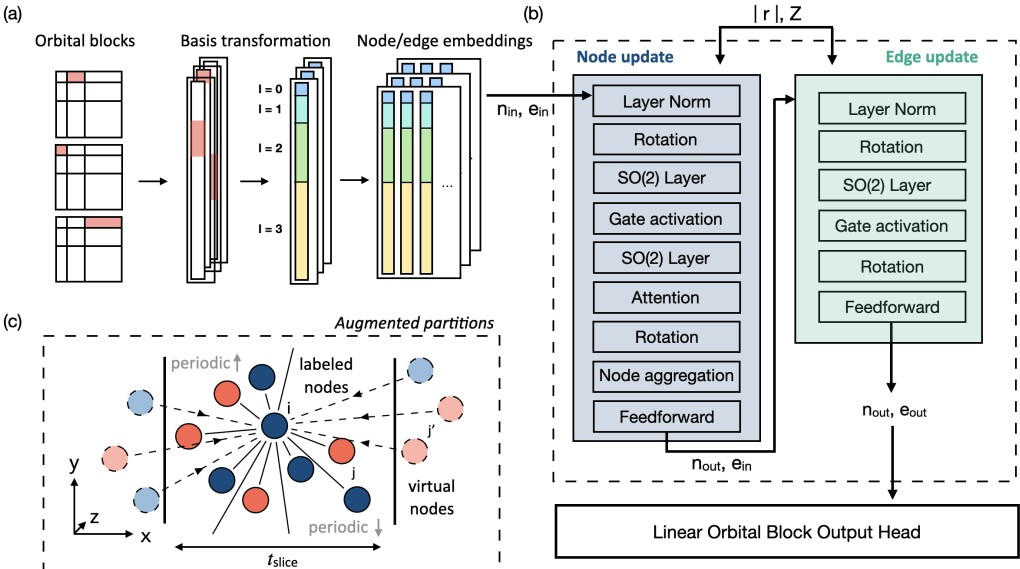

*Figure 3.* **(a)** Data transformation of the Hamiltonian matrix. Blocks of orbital interactions are first extracted from $\boldsymbol{H}$ and reshaped into input tensors, which are transformed into the coupled basis. The tensor corresponding to each node and edge is expanded into a dimension $E_n/E_e$, and this set of initial embeddings for every node and edge is sent into **(b)** the node update block. The node features ($\boldsymbol{n}_X$) are updated using a message-passing scheme. The edge features ($\boldsymbol{e}_X$) are updated based on the learned $\boldsymbol{n}_X$. Z refers to atomic numbers, while $\mathrm{r}_{ij}$ is the set of scalar distances between atoms. **(c)** Illustration of how connectivity beyond the partition boundaries is incorporated through the addition of virtual edges ($e_{j'\rightarrow i}$) from virtual nodes ($j'$) to the labeled nodes. A set of edges from labeled and virtual nodes is shown for a single node ($i$) within the partition. Solid vertical lines indicate the partition boundaries, and different colors represent different atomic species.

inability to divide the graph into batches also slows down the training process overall.

To efficiently train graphs corresponding to large materials while maintaining correct atomic environments and neighboring edge connections, we introduce an *augmented partitioning* approach. A visual representation of it is provided in **Fig. 3(c)**. The graph is first partitioned into subgraphs where the data can fit into the memory of a single GPU. Atoms located outside of a given partition, but connected to those within, are represented by virtual nodes (**Fig. 3(c)** - dashed circles). They are attached to the partition through virtual edges (**Fig. 3(c)** - dashed lines). These virtual nodes/edges are initialized similarly to their labeled counterparts with input atomic numbers and distances. However, their outputs are not used. Their only purpose is to inform each partition of its full connectivity so that the network can then learn an accurate and generalizable aggregation function during message passing. To leverage the periodic boundaries of the material structures treated here, we partition the input structures by dividing the graph into 'slices' along the longest dimension ($x$-axis), retaining edges across the $y$- and $z$-boundaries. Details about the construction of partitions are provided in **Appendix B**.

As the set of virtual connections used to augment each graph includes only the 1-hop neighborhood, the network is strictly local. Limiting the receptive field is an approach often used to increase scalability. As demonstrated by previous strictly local architectures, e.g., Allegro (Musaelian et al., 2023), information from the local environment is often sufficient to achieve state-of-the-art prediction accuracy when interactions are strongly localized and the interaction cutoff is sufficiently large. In **Appendix F**, we present further details on how the Hamiltonian matrix elements satisfy this locality. To capture higher body-order information, many-body interactions can be flexibly added into the network without increasing the receptive field, as implemented by (Zhouyin et al., 2024).

### 3.4. Dataset creation

To generate sufficiently rich training data, existing datasets typically sample molecules at various time steps of molecular dynamics (MD) trajectories (Yu et al., 2023a; Schütt et al., 2019; Christensen & von Lilienfeld, 2020) or generate multiple small perturbations of the atoms in a crystalline lattice (Li et al., 2022). Here, as a representative subset of realistic (disordered) materials, we consider amorphous crystals and take advantage of the fact that (1) every node

| Material | Structure | Purpose | $r_{cut}$ [Å] | # atoms | # orbitals | # edges | $x$ [Å] | $y$ [Å] | $z$ [Å] |
|----------|-----------|---------|---------------|---------|------------|---------|---------|---------|---------|
| a-HfO$_2$ | 1 | validate | 8 | 3,000 | 18,000 | 527,348 | 52.876 | 26.308 | 26.242 |
| a-HfO$_2$ | 2 | train | 8 | 3,000 | 18,000 | 533,364 | 52.346 | 26.237 | 26.293 |
| a-HfO$_2$ | 3 | test | 8 | 3,000 | 18,000 | 530,920 | 52.722 | 26.267 | 26.191 |
| a-HfO$_2$ | 3 | test | 12 | 3,000 | 18,000 | 1,792,760 | 52.722 | 26.267 | 26.191 |
| a-GST | 1 | train/validate (6:1 split) | 12 | 1,008 | 13,104 | 230,848 | 29.541 | 25.583 | 41.777 |
| a-GST | 2 | test | 12 | 1,008 | 13,104 | 226,406 | 25.857 | 29.857 | 41.691 |
| a-PtGe | 1 | train/validate (10:1 split) | 8 | 2,688 | 16,128 | 319,262 | 82.283 | 23.171 | 25.031 |
| a-PtGe | 2 | test | 8 | 2,688 | 16,128 | 319,306 | 82.283 | 23.171 | 25.031 |
| a-PtGe | 2 | test | 10 | 2,688 | 16,128 | 629,790 | 82.283 | 23.171 | 25.031 |

*Table 1.* Attributes of the generated dataset for three materials, each with its own training, validation, and test set: The $[x, y, z]$ triplet defines the periodic unit cell size. $nnz_H$ is the number of non-zero elements in the Hamiltonian, encompassing all orbital interactions. Edges were defined according to an interaction distance $r_{cut}$.

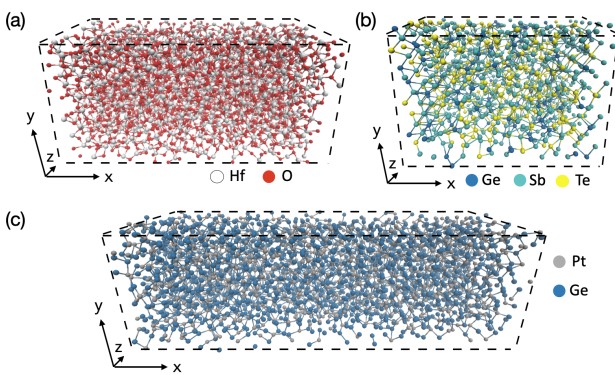

*Figure 4.* Atomic structure example for the three materials we consider. In each case, the black dashed box illustrates the boundaries of the repeating unit cell.

has a different local atomic environment, and (2) each structure contains a large sampling of different motifs. A wide range of training data can thus be captured within a single sample.

We generate a custom dataset for three different materials in the amorphous (a-) phase (without long-range order): a-HfO$_2$, a-GeSbTe (a-GST), and a-PtGe. Besides exhibiting features that require large unit cells to be described, these materials are also scientifically and technologically relevant: a-HfO$_2$ is a common high-$\kappa$ dielectric that can be found in almost all transistors and integrated circuits (Chan et al., 2008). a-GST and alloys of a-Ge show high resistance contrasts between crystalline and amorphous phases, and can be used as switching layers of non-volatile memory cells (Pirovano et al., 2004; Kolobov et al., 2004; Zellweger et al., 2024) that are fully compatible with CMOS fabrication processes. Knowing the electronic Hamiltonian of these materials is essential to understanding the behavior of the corresponding devices and to designing better-performing

components (Brandbyge et al., 2002). Structure examples can be visualized in **Fig. 4** for each material. The corresponding structural information used for training, validation, and testing is found in Table 1. Details behind the sample generation are given in **Appendix D**. This dataset will be made publicly available to serve as a reference for large electronic Hamiltonian matrix predictions.

# 4. Results

For fair comparisons in experiments where the quantity of training data may vary, we used a ReduceLRonPlateau scheduler that reduces the learning rate when no further decrease in validation loss is detected. Training is stopped once a minimum learning rate is reached. The values of the hyperparameters and the scheduler settings for different experiments are discussed in **Appendix C**.

## 4.1. Ablation studies of the training approach

We study the model's ability to generalize to different configurations of large systems by predicting the Hamiltonian matrix $H$ of a full a-HfO$_2$ sample (structure 3 in **Table 1**), which remains unseen during the training process. For all subsequent HfO$_2$ experiments, the *augmented partitioning* scheme is only applied during training on structure 2, while the $H$ of the full unseen structure is predicted during inference (structure 3). We use an r$_{cut}$ of 8 Å. Errors are reported separately for nodes ($\epsilon_n$) and edges ($\epsilon_e$) to distinguish intra- and inter-atomic orbital interactions, which typically have very different magnitudes.

First, we demonstrate the improvement in accuracy resulting from the *augmented partitioning* approach introduced in **Section 3.3**. In particular, we examine the influence of virtual nodes/edges in **Table 2**. Compared to training with raw partitions, the addition of virtual nodes and edges reduces the node ($\epsilon_{node}$) and edge prediction error ($\epsilon_{edge}$) by $\sim$60% and $\sim$70% respectively. Such an improvement is expected,

|  | $\epsilon_{\mathbf{n}}[mE_h]$ | $\epsilon_{\mathbf{e}}[mE_h]$ |
|---|---|---|
| $n' \quad n$ | 5.39 | 0.55 |
| $n' \to n$ | 2.16 | 0.16 |

*Table 2.* Ablation study on the impact of virtual nodes on the prediction accuracy. Testing is done on structure 3, using slices of length $t_{slice} = 3$ Å from structure 2 for training and of length $t_{slice} = 4$ Å from structure 1 for validation. The first column indicates the connectivity between virtual ($n'$) and labeled ($n$) nodes. $\epsilon_n/\epsilon_e$ is the error for nodes/edges.

as raw partitions omit a large proportion of boundary edges and thus incorrectly capture atomic neighborhoods. We note that as $t_{slice} < r_{cut}$, the one-hop neighborhood connects all nodes within each partition.

Next, we explore the impact of augmented partitioning on accuracy by training on an increasingly sub-divided graph in **Table 3**. The total number of labeled atoms used for training remains the same (3,000), while the total number of labeled edges decreases with more partitions. Despite the different divisions ranging from 5 ($t_{slice} \simeq 12$ Å) to 27 ($t_{slice} \simeq 2$ Å) slices, $\epsilon_{\mathbf{n}}$ and $\epsilon_{\mathbf{e}}$ remain very close to the values obtained by training with the full graph ($t_{slice}$ = 52.346 Å). The prediction error is thus insensitive to partition size. For small slices in this case, the reduced fraction of labeled connections along the $x$ direction does not affect the accuracy, as the remaining data along the $y$ and $z$ directions is sufficient to train the network.

| $t_{slice}$ [Å] | $N_t$ | $N_e$ | Epochs | $\epsilon_{\mathbf{n}}[mE_h]$ | $\epsilon_{\mathbf{e}}[mE_h]$ |
|---|---|---|---|---|---|
| ~2 | 27 | 95,398 | 12,143 | 2.43 | 0.23 |
| ~3 | 18 | 141,512 | 17,736 | 2.16 | 0.16 |
| ~4 | 14 | 184,730 | 17,726 | 2.24 | 0.16 |
| ~8 | 7 | 320,324 | 13,480 | 2.64 | 0.20 |
| ~12 | 5 | 381,504 | 14,251 | 2.75 | 0.19 |
| ~52 | 1 | 533,364 | 17,528 | 2.50 | 0.17 |

*Table 3.* Prediction accuracy for HfO$_2$ when the network is trained on differently-sized partitions of the same graph (structure 2), using 1 MP layer. $N_t$ is the number of slices, $N_e$ is the total number of labeled edges. The total number of labeled nodes remains constant. The number of slices is equal to $\sim L/t$, where $L$ = 52.346 Å is the full length of structure 2. The validation set is a slice of length $t_{slice} = 4$ Å starting at $x_0 = 25$ Å from structure 1. The models are tested on the full unseen structure 3.

## 4.2. Structural disorder

We apply the same local architecture to our custom dataset of structurally disordered (amorphous) materials (HfO$_2$, GST, and PtGe from **Fig. 4**/**Table 1**), which contain a range of different atomic elements (Hf, O, Ge, Sb, Te, Pt), basis sets (SZV, DZVP), and bonding behavior (ionic, covalent). All models are trained using augmented partitions and tested

on full unseen structures belonging to the same material. The results are summarized in **Table 4**.

The strictly local architecture trained on augmented partitions performs consistently across the different test datasets. In all cases, prediction errors remain relatively insensitive to the partition size (**Table 11** in **Appendix I**). Partitioning also allows us to extend $r_{cut}$ further (to 12 Å for HfO$_2$) while remaining within GPU memory limitations, effectively encompassing all non-zero matrix elements. As a result, the best achievable error for each material system ranges from 2.17 meV to 2.58 meV, for test structures containing 1000-3000 atoms and 200,000+ to 1,792,760+ edges. These values are comparable to what a previous study obtained (2.2 $meV$) using equivariant GNNs for smaller structures with $\leq$150 atoms per unit cell (Wang et al., 2024b).

## 4.3. Compositional disorder

a-HfO$_2$ often exists in a sub-stoichiometric form (a-HfO$_x$) due to the presence of oxygen vacancies induced by the growth process. The distribution of these vacancies varies significantly from one sample to the other, introducing a statistical component to computational studies of the HfO$_x$ electronic properties. Accounting for these compositional variations necessitates a distinct DFT simulation for each stoichiometric different structure, which is computationally intensive, but an ideal application of machine learning solutions.

To demonstrate our network's ability to treat such problems, we train a model on a single sub-stoichiometric a-HfO$_{1.8}$ structure where 10% of the oxygen atoms were replaced by randomly distributed vacancies represented as ghost atoms. We then use it to predict unseen full structures with a stoichiometry of a-HfO$_{1.7}$, a-HfO$_{1.8}$, and a-HfO$_{1.9}$. The $\epsilon_n$ and $\epsilon_e$ (**Table 5**) remain within a small range (2.34-2.45 $mE_h$ and 0.15-0.16 $mE_h$, respectively), regardless of the vacancy concentration and distribution, showing that the network generalizes very well to compositional disorder, despite being trained on only one configuration. Models trained on other vacancy concentrations perform similarly. Their results, along with details of the vacancy implementation, can be found in **Appendix I.1**.

## 4.4. Eigenvalue spectrum of predicted a-HfO$_2$

Next, we assess whether the prediction accuracy of the trained network is sufficient for practical application. For this, we use HfO$_2$, as it has the highest error in our experiments in **Table 4** and thus presents an upper bound on the expected accuracy. We assemble the full Hamiltonian of the HfO$_2$ test structure using the network outputs ($\boldsymbol{H}^{pred}$), trained with 18 partitions of length $t_{slice} = \sim 3$ Å. We then compute the eigenvalue spectrum of $\boldsymbol{H}^{pred}$ and its reference $\boldsymbol{H}^{GT}$, as well as the error distribution between them

| Material | $t_{slice}[\text{Å}]$ | $N_t$ | $r_{cut}[\text{Å}]$ | $N_n$ | $N_e$ | $\epsilon_{\mathbf{n}}[mE_h]$ | $\epsilon_{\mathbf{e}}[mE_h]$ | $\epsilon_{\mathbf{tot}}[mE_h]$ | $\epsilon_{\mathbf{tot}}[meV]$ |
|---|---|---|---|---|---|---|---|---|---|
| a-GST | 5 | 6 | 12 | 1,008 | 226,406 | 0.97 | 0.08 | 0.09 | 2.40 |
| a-PtGe | 5 | 10 | 8 | 2,688 | 319,306 | 0.77 | 0.08 | 0.09 | 2.49 |
| a-PtGe | 5 | 10 | 10 | 2,688 | 629,790 | 0.80 | 0.08 | 0.08 | 2.17 |
| a-HfO$_2$ | 3 | 18 | 8 | 3,000 | 530,920 | 2.16 | 0.16 | 0.18 | 4.84 |
| a-HfO$_2$ | 3 | 18 | 12 | 3,000 | 1,792,760 | 2.13 | 0.09 | 0.10 | 2.58 |

*Table 4.* Summary of model performance when trained on amorphous GST, PtGe, and HfO$_2$. $N_n$ and $N_e$ refer to the number of nodes and edges of the test structure. respectively. Key parameters of the structures used for training, validation, and testing are given in Table 1.

| Stoichiometry ($x$) | | $\epsilon_{\mathbf{n}}[mE_h]$ | $\epsilon_{\mathbf{e}}[mE_h]$ |
|---|---|---|---|
| Train set | Test set | | |
| 1.8 | 1.9 | 2.34 | 0.15 |
| 1.8 | 1.8 | 2.38 | 0.15 |
| 1.8 | 1.7 | 2.45 | 0.16 |

*Table 5.* Model trained on a-HfO$_{x=1.8}$ and tested on full unseen structures with different stoichiometry ($x$). 18 slices of each structure, each 3 Å thick, are used for training. The training method is identical to the one from Table 3.

(**Fig. 5**(a)). The predicted $\boldsymbol{H}^{pred}$ is able to reproduce all eigenvalues of $\boldsymbol{H}^{GT}$ within 0.530% relative L1 error. The error decreases to 0.446% when eigenvalues of unoccupied states (those situated above 0.306 $E_h$) are excluded. The remaining error is carried mostly by the largest eigenvalues and distributed around the edges of energy gaps (see **Fig. 9** in **Appendix F**), which correspond to regimes of stronger inter-atomic orbital coupling (Atkins & De Paula, 2009).

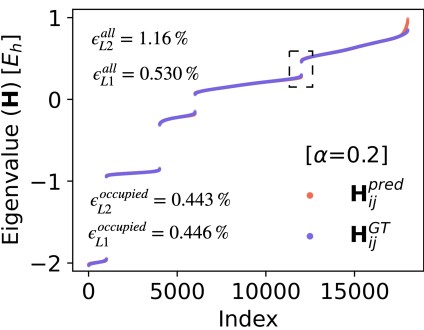

*Figure 5.* (a) Eigenvalue spectrum of the predicted ($\boldsymbol{H}^{pred}$) and reference ($\boldsymbol{H}^{GT}$) Hamiltonian matrices. The alpha value indicates the scatter point transparency. $(\boldsymbol{H}_{i,j})^{pred}$ is symmetrized before diagonalization with $\boldsymbol{H} = \frac{1}{2}(\boldsymbol{H} + \boldsymbol{H}^{\dagger})$. The relative L1/L2 errors in the eigenvalue spectrum, computed as $(\|(\vec{E})^{pred} - (\vec{E})^{GT}\|^2_{norm})/(\|(\vec{E})^{GT}\|^2_{norm})$ (norm = 1, 2) where $\vec{E}$ is the vector of eigenvalues, are shown for all eigenvalues and for the ones corresponding only to occupied states (below 0.305 $E_h$). The black dashed box indicates the bandgap.

### 4.5. Computational cost

Compared to full-batch training of the graph, our method using 8 augmented slices results in a 6.5× speedup per epoch (0.38 vs. 2.5 s) and a 7.2× decrease in memory consumption per rank (8.59 vs. 61.68 GiB) without affecting accuracy. A more complete analysis is provided in **Appendix H.2**. This scaling behavior is only limited by the overhead introduced to process the virtual nodes/edges and by any computational load imbalance from partitioning. Further computational improvements could be achieved by combining the augmentation approach with optimized graph partitioning algorithms extended to leverage periodicity.

The extension of GNN-based predictions to large material systems could potentially save tremendous amounts of computational time, as the inference phase scales with $\mathcal{O}(N_{atoms})$ while DFT calculations are limited to $\mathcal{O}(N_{atoms}^3)$. While DFT calculations to obtain the $\boldsymbol{H}^{GT}$ of small molecules (e.g., H$_2$O) take only a few seconds, the same operation for a-HfO$_2$ structures made of 3,000 atoms is computationally two orders of magnitude heavier (0.04 vs. 3.65 node hours, see **Appendix H**). We have thus demonstrated the applicability of GNN approaches to a regime where exact solutions are almost entirely prohibitive for downstream applications. The model could also serve as an initial guess to DFT packages to reduce the number of self-consistent field iterations that are required to obtain converged electron densities (Unke et al., 2021).

## 5. Applications in Device Transport Modeling

Investigating materials at the device scale is one of the next challenges to tackle for ML-based materials modeling (Miret et al., 2025). Here we demonstrate that Hamiltonian matrices predicted by our approach can serve as inputs to *ab-initio* quantum transport simulations, which aim to compute the electrical current flowing through materials under an applied voltage bias. This is done through the solution of the Schrödinger equation with open boundary conditions, where the Hamiltonian matrix contains the electronic properties of the treated device. From the produced energy-resolved transmission function, the electrical current can be derived via the Landauer-Büttiker formula (Brandbyge et al., 2002).

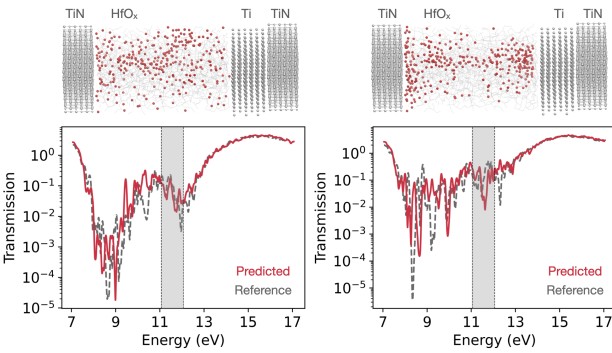

*Figure 6.* TiN/a-HfO$_2$/Ti/TiN structures with different oxygen vacancy distributions (top), which lead to distinct electrical transmission curves (bottom). The positions of vacant oxygen atoms in the former are indicated with red points. The grey boxes in the transmission plots indicate the difference in the Fermi levels of the two contacts, created through an applied bias. Current flow is determined primarily by the transmission through this energy window.

As a test, we designed a typical metal-insulator-metal resistive switching device by embedding the a-HfO$_x$ structure in **Fig. 4** with two metal contacts and creating a TiN/a-HfO$_2$/Ti/TiN stack (Kaniselvan et al., 2023). We then computed its electrical current for a-HfO$_x$ layers with different vacancy configurations. The electrical transparency of a-HfO$_x$ is a function of its exact atomic structure as well as its local stoichiometric variations. Initially, the stoichiometric a-HfO$_2$ blocks current flow due to its finite band gap. The gradual introduction of oxygen vacancies changes this stoichiometry and introduces energy states within the band gap which can carry current (Devices 1-2). This current is further enhanced when defects form a filament bridging the metallic electrodes.

We modeled this process using either a (1) DFT- or (2) ML-generated Hamiltonian. The resulting transmission functions and electrical current values are presented for two TiN/a-HfO$_x$/Ti/TiN structures in **Fig. 6** and **Table 6**, respectively. Training for the latter occurred on slices of amorphous HfO$_x$ structures with a uniform distribution of oxygen vacancies. In the test structures, these oxygen vacancies have formed partial or complete conductive filaments. We thus test whether the learned electronic structures of isolated vacancies can be extended to the case of tight interactions between them, which are responsible for the current.

While the quantitative agreement between the transmission functions and electrical currents obtained with a DFT and ML Hamiltonian is not perfect, it is sufficient to (1) track the difference in current across five orders of magnitude, (2) match physically relevant features such as the band edges of the a-HfO$_x$ layer or localized regions with higher transmission, and (3) capture the qualitative difference in elec-

trical transparency caused by different spatial distributions of oxygen vacancies. Device simulations with a ML-based Hamiltonian can therefore be used to understand the connection between atomic structure and electrical current flow in TiN/a-HfO$_x$/Ti/TiN stacks. Under this level of accuracy, the model can now also generate Hamiltonian matrices for systems with sizes which are otherwise computationally infeasible with DFT.

| Device | $\epsilon_{\mathbf{n}}[mE_h]$ | $\epsilon_{\mathbf{e}}[mE_h]$ | $I_{ref}$ [A] | $I_{pred}$ [A] |
|---|---|---|---|---|
| 0 | 1.71 | 0.16 | $8.00 \times 10^{-9}$ | $5.54 \times 10^{-9}$ |
| 1 | 1.66 | 0.16 | $1.48 \times 10^{-5}$ | $1.06 \times 10^{-5}$ |
| 2 | 1.59 | 0.15 | $6.99 \times 10^{-6}$ | $4.89 \times 10^{-6}$ |

*Table 6.* Summary of prediction results and computed currents for devices with vacancy configurations forming different filament shapes. Devices 1-2 are as in **Fig. 6**.

## 6. Conclusion

We developed an equivariant GNN tailored to learn the electronic properties of large-scale, disordered materials, and introduced an *augmented partitioning* approach to break down and train the graphs encountered when dealing with realistic materials. More generally, we proposed a method to tackle the training of systems represented by large, highly connected, and near-sighted graphs where a strictly local environment is sufficient. Our approach can be straightforwardly applied to other complex materials, or adapted to learn their other rotationally-equivariant attributes, such as vibrational properties, e.g., phonon dispersions (Fang et al., 2024). The resulting network captures relevant properties of large structures in sufficient detail to achieve few-$meV$ accuracy and reproduce the energy eigenvalues to under 0.6% error. Further data generation, network optimization, and enabling increased expressiveness will be the next steps.

## Acknowledgements

We acknowledge funding from the ALMOND project (SNSF Sinergia grant no. 198612) and from the Swiss State Secretariat for Education, Research, and Innovation (SERI) through the SwissChips research project. This work was also supported by the Swiss National Supercomputing Center (CSCS) under project lp16. We would also like to thank Paul Uriarte Vicandi and Luiz Felipe Aguinsky for providing the HfO$_2$ and PtGe structures, respectively.

## Impact Statement

The goal of this work is to advance research in computational materials science through machine learning. There are many potential societal consequences of our work, none of which we feel must be specifically highlighted here.

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

2407.06053.

## A. Loss computation during training and inference

For all experiments, a minor difference from the procedures reported in (Yu et al., 2023b) and (Schütt et al., 2019) is that we use the Mean Squared Error (MSE) of the full output and target vectors in the coupled space to compute the fitting and validation loss during training: $\mathcal{L}_{MSE}(x_i, \hat{x}_i) = \frac{1}{N} \sum_{i=1}^{N} (x_i - \hat{x}_i)^2$, where $\boldsymbol{x}$ and $\hat{\boldsymbol{x}}$ are the flattened targets in the coupled space. These targets are padded with zeros to ensure that those of different orbital interactions have the same dimensions. We note that doing so burdens the network to learn the zero-padding in addition to the data, which can artificially increase the number of epochs required for training. However, the computational cost of extracting padding-free data and post-processing it at the end of each epoch is also non-negligible. For this work we therefore use the padded data during training for ease of implementation. The final reported loss for all our results uses the Mean Absolute Error (MAE) after converting the output and label tensors back into uncoupled space and extracting the un-padded data: $\mathcal{L}_{MAE}(x_i, \hat{x}_i) = \frac{1}{N_{orb}} \sum_{i=1}^{N_{orb}} |x_i - \hat{x}_i|$

## B. Augmenting graph partitions with virtual nodes

In **Algorithm 1**, we detail the procedure to partition the full graph $\mathcal{G}$, described by the set of vertices $\mathcal{V}$ and edges $\mathcal{E}$, into a set of slices $\{\mathcal{G}_1 \ \ldots \ \mathcal{G}_N\}$ which are augmented by virtual nodes and edges. The number of labeled nodes ($n_i^n$) and the number of labeled edges ($n_i^e$) are collected and passed to the training functions, which then omit the remainder of the outputs (the virtual nodes and edge outputs) while computing the loss.

---

**Algorithm 1:** Augmented partitioning approach

---

1   Graph $\mathcal{G}(\mathcal{V}, \mathcal{E})$ Slice x-coordinates $[x_1 \ \ldots \ x_N]$ Set of subgraphs $\{\mathcal{G}_1 \ \ldots \ \mathcal{G}_N\}$ Numbers labeled nodes $[n_1^n \ \ldots \ n_N^n]$ Numbers labeled edges $[n_1^e \ \ldots \ n_N^e]$

2   **for** $i \leftarrow 1$ **to** $N$ **do**
3      $\mathcal{V}_i \leftarrow []$;
4      $n_i^n = 0$;
5      **for** $v \in \mathcal{V}$ **do**
6          **if** $v.x \in [x_i, x_{i+1})$ **then**
7             $\mathcal{V}_i$.append(v);
8             $n_i^n \mathrel{+}= 1$;
9          **end**
10      **end**
11      $\mathcal{E}_i \leftarrow []$;
12      $n_i^e = 0$;
13      **for** $v_1 \in \mathcal{V}_i$ **do**
14          **for** $v_2 \in \mathcal{V}_i$ **do**
15             **if** $v_1 \rightarrow v_2 \in \mathcal{E}$ **then**
16                 $\mathcal{E}_i$.append($v_1 \rightarrow v_2$);
17                 $n_i^e \mathrel{+}= 1$;
18             **end**
19          **end**
20      **end**
21      **for** $v_1 \in \mathcal{V}_i$ **do**
22          **for** $v_2 \in \mathcal{V} \setminus \mathcal{V}_i$ **do**
23             **if** $v_2 \rightarrow v_1 \in \mathcal{E}$ **then**
24                 $\mathcal{V}_i$.append($v_2$);
25                 $\mathcal{E}_i$.append($v_2 \rightarrow v_1$);
26             **end**
27          **end**
28      **end**
29      $\mathcal{G}_i(\mathcal{V}_i, \mathcal{E}_i)$;
30   **end**

---

The augmentation approach can be combined with any method of partitioning the input graph. An ideal partitioning scheme would result in the maximal ratio of labeled nodes/edges within each sub-graph, compared to virtual nodes/edges. As we consider structures with fully periodic boundaries in all three dimensions, a simple heuristic to leverage this periodicity is to partition along only one dimension ($x$), thus maintaining all the labeled edges across the $y-$ and $z-$ cell boundaries of each periodic image. This motivates the division by slices described in **Algorithm 1**, which can be very effective when assuming a constant atomic density. Finding more optimal partitions that nevertheless leverage periodicity is a subject of future work.

## C. Hyperparameters

We use the hyperparameters shown in Table 7 to train on all datasets. The ReduceLRonPlaeau scheduler decreases the learning rate by the decay factor when it does not detect a further decrease in validation loss within the decay patience $t_{patience}$. The threshold refers to the sensitivity of the scheduler to changes in validation loss. Once the minimum learning rate is reached, the training stops.

| Hyper-parameters | a-HfO$_2$/PtGe/GST dataset |
|---|---|
| Optimizer | Adam |
| Precision | single (f32) |
| Scheduler | ReduceLROnPlateau |
| Initial learning rate | $1 \times 10^{-4}$ |
| Minimum learning rate | $1 \times 10^{-5}$ |
| Decay patience $t_{patience}$ | 500 |
| Decay factor | 0.5 |
| Threshold | $1 \times 10^{-3}$ |
| Maximum degree $L_{max}$ | 4 |
| Maximum order $M_{max}$ | 4 |
| Embedding size | 16 |
| Number of attention heads $N_h$ | 2 |
| Feedforward Network Dimension | 64 |

*Table 7.* Hyper-parameters used for a-HfO2, a-PtGe and a-GST data.

## D. Dataset generation

Atomic structures corresponding to materials in the amorphous phase are not straightforward to generate since they must accurately capture the structural motifs underlying this phase and a realistic range of atomic coordination. To accurately reproduce long-range structural disorder, the structures used must also be large enough to avoid the creation of wavefunctions that repeat over periodic boundaries. Existing methods to do so include melt-quench (Urquiza et al., 2021), seed-and-coordinate (Youn et al., 2014), or 'decorate and relax' (Tafen & Drabold, 2003) approaches.

In this work, we use melt-quench processes with molecular dynamics (MD) to evolve each of the three materials considered from their crystalline phases, following a similar procedure as the ones described in (Kaniselvan et al., 2023) and (Urquiza et al., 2021). We then perform a structural relaxation with CP2K code (Kühne et al., 2020) to correct for any discrepancies between the relaxed bond lengths attained with the force field used for MD and those obtained with DFT. Due to the large cell sizes of the a-HfO$_2$ structures, all necessary information is contained within the $\Gamma$ point (where the wavevectors $k_x = k_y = k_z = 0$). The energies at this location can be computed by directly diagonalizing $\boldsymbol{H}$. The datasets are publicly available at https://huggingface.co/datasets/chexia8/Amorphous-Hamiltonians.

Further details specific to each material are provided in the sections below.

### D.1. a-HfO$_2$

We generate 3 independent structures of a-HfO$_2$ using the QuantumATK toolkit (Søren Smidstrup et al., 2020). As a first step, we run an MD NVT simulation at 3000K for 50 ps with a step size of 1 fs. We use the MTP-HfO$_2$-2022 potential provided by the software. Next, we run an NPT simulation for 300 ps (and the same 1 fs step size), with an initial reservoir

temperature of 3000K and a final temperature of 300K, for a cooling rate of 9K/ps. Finally, we anneal the structure at 300K for 50 ps.

Due to the computational cost of using a more complete Double-$\zeta$ Valence Polarized (DZVP) basis set (VandeVondele & Hutter, 2007), we use a simpler Single-$\zeta$ Valence (SZV) basis (VandeVondele & Hutter, 2007), which uses 4 basis functions per Oxygen atom and 10 basis functions per Hafnium atom. The plane-wave cutoff is set to 500 Ry, while a cutoff of 60 Ry is used for mapping the Gaussian-type orbitals onto the grid. We use the PBE functional for the exchange-correlation energy (Perdew et al., 1997). To accurately capture the band gap of a-$HfO_2$, we apply the Hubbard correction (Anisimov et al., 1991) of U = 7 eV to the 3d orbital of Ti and the Hubbard correction of U = 10 eV to the 2p orbital of O.

### D.2. Substoichiometric $HfO_x$

We create a dataset for sub-stoichiometric $HfO_x$ structures by introducing randomly distributed oxygen vacancies into the original, pristine $HfO_2$ structures. The sub-stoichiometric structures are generated for x = 1.9, 1.8, and 1.7 (corresponding to vacancy concentrations of 5%, 10%, and 15 %, respectively). Vacancies are treated as ghost atoms (atoms with no orbitals, but with a basis set defined at their locations), to mitigate the basis set superposition error (Senent & Wilson, 2001), a known problem related to localized basis sets. More precisely, by treating vacancies as ghost atoms, one prevents the excessive borrowing of the basis sets from neighboring atoms by the vacancy, which improves the accuracy of the predicted electronic properties. These ghost atoms are assigned an atomic number of 0.

### D.3. a-GST

We use two amorphous GST-124 ($Ge(SbTe_2)_2$) structures containing 1008 atoms for training and validation. The first structure is extracted from the MD simulation of GST-124 crystallization, provided by (Yuxing Zhou, 2023) (Supplementary Material). The initial geometry for the second structure is a perfectly crystalline GST-124 structure, taken from Materials Project database (Jain et al., 2013). The amorphous structure is then generated through a standard melt-quench procedure, consisting of atomic position randomization at 3000K for 20 ps, cooling to the melting point of 600K at a rate of $10^{14} K.s^{-1}$, holding for 30 ps, quenching to 300K at a rate of $2.5 \times 10^{13} K.s^{-1}$, and finally, annealing at 300K for 50 ps. Both amorphous structures are obtained via MD simulations in LAMMPS (Thompson et al., 2022), equipped with the QUIP library for Gaussian Approximation Potential (GAP) (Csányi et al., 2007). The corresponding Hamiltonian matrices are obtained using CP2K, where we run calculations with the DZVP basis, the plane-wave cutoff of 300 Ry, the Gaussian-type orbitals mapping cutoff of 50 Ry, and the PBE functional.

### D.4. a-PtGe

To generate the PtGe structures, germanium structures are taken from the Materials Project database (Jain et al., 2013). This is followed by an NVT melt-quench process using LAMMPS and Stillinger-Weber parameters (Stillinger & Weber, 1985; Nordlund et al., 1998; Wang & Stroud, 1988). The structures are heated to a melting temperature of 5000K at a rate of 0.47 $10^{12} K.s^{-1}$, kept at the melting temperature for 20000 ps (structure 1) or 22000 ps (structure 2), quenched at a rate of 4.7 $10^{12} K.s^{-1}$, and finally annealed at 300K for 100 ps. Using the Atomic Simulation Environment (ASE) tool (Hjorth Larsen et al., 2017), 1/3 of the Ge atoms are replaced by Pt atoms. The cell of the alloy is then stretched to match the cell of a $PtGe_2$ structure (taken from the Materials Project and optimized using CP2K). Fixed-volume geometry relaxation is then performed on the PtGe alloy. For the structural optimization, as well as for the $H$ and $S$ generation, SZV basis set and PBE exchange-correlation functionals are used. We apply a plane-wave cutoff of 1000 Ry and a cutoff for Gaussian-type orbitals mapping of 70 Ry.

## E. Atomic bonding environments in the amorphous phase

We use the example of a-$HfO_2$ to investigate the structural differences between samples generated from different starting melt-quench processes. In **Fig. 7** we plot the O-coordination of each Hf atom and the radial distribution function $g(r)$ (where r is the inter-atomic distance) for each of the three structures. The distribution in the coordination and dispersion of the peaks in $g(r)$ indicates the amorphous nature of the three structures. Variations between them appear as perturbations in these two quantities. To gain more insights into how different the structures are, we additionally plot the spatially resolved O-coordination of Hf atoms along the longest, $x$ coordinate for the three structures, as well as the distributions of outliers (Hf atoms with very low and very high O-coordination) in three-dimensional space. These outliers are situated at different

locations in different structures, demonstrating a significant degree of dissimilarity among the structures.

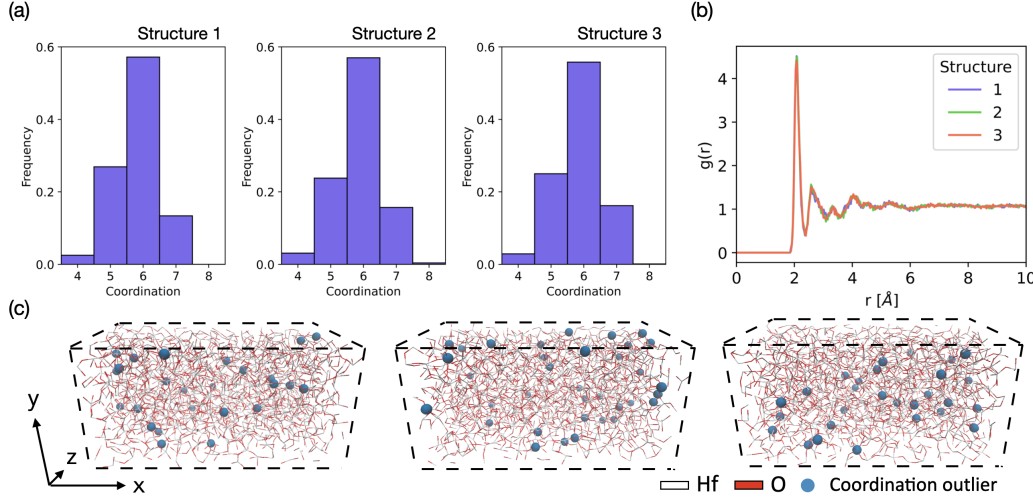

*Figure 7.* a) O-coordination of Hf atoms (number of O atoms bonding a Hf atom) for each of the a-HfO₂ structures, showing a distribution around a coordination number of 6, and variation between the structures. b) The radial distribution function ($g(r) = \frac{dn(r)}{dr} \frac{V_{domain}}{4\pi r^2 N_{atoms}}$), where $n(r)$ is the number of atoms with distance $r$ between them for the structures 1-3. (c) Spatial distribution of coordination outliers (Hf atoms with O-coordination equal to 8 or 4) for the three structures, which are an indicator of the uniqueness of the three structures.

## F. Near-sightedness of the Hamiltonian matrix

In Fig. 8, we visualize the matrix elements corresponding to the Hamiltonian of one sample of each material from the dataset. Specifically, we show the interaction ranges at which non-zero matrix elements exist, as well as their decay with increasing interatomic distance. When using a local basis, this near-sightedness holds strictly.

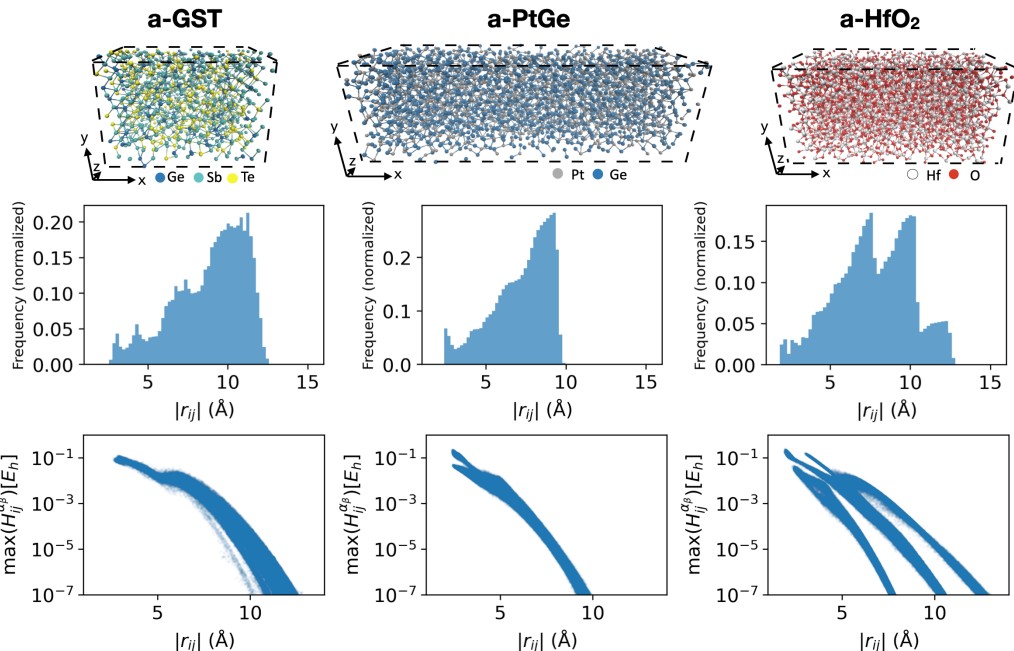

*Figure 8.* Summary of the Hamiltonian matrix properties for each of the materials in our custom dataset. Shown are (top row) atomic structures, (middle row) frequency of matrix elements as a function of interatomic distances, and (bottom row) distribution of the maximum element of each block of $\boldsymbol{H}$ as a function of interatomic distance $|r_{ij}|$.

Next, using the three a-HfO$_2$ structures in **Table 1** as an example, we show the distribution of energy eigenvalues in **Fig. 9** at different values of $r_{cut}$. In the second row, we zoom into the range of eigenvalues around the energy bandgap, which is defined by the transition between occupied and unoccupied electronic states (the Fermi level $E_F = \sim 0.3 E_h$ in all cases). Values of $r_{cut} \geq 8$Å create no noticeable difference on the eigenvalue spectra. Note that the value of $r_{cut} = \infty$ corresponds to the case where no nonzero values were filtered from $\boldsymbol{H}^{GT}$.

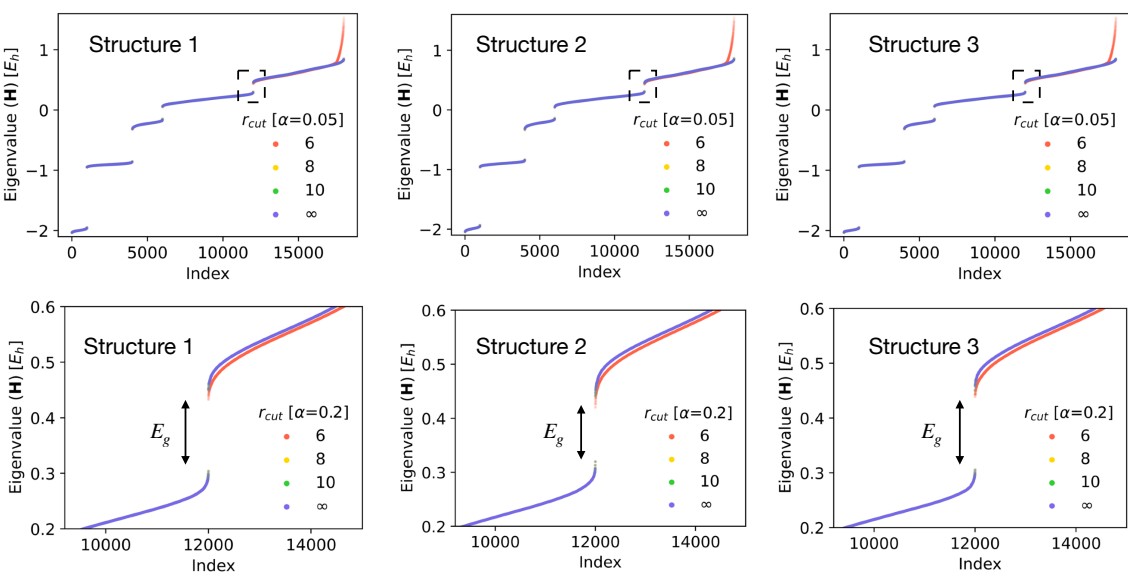

*Figure 9.* Eigenvalues of the ground-truth Hamiltonian matrix, showing (top) the full eigenvalue spectrum and (bottom) the spectrum around the bandgap, for the three structures, in the order of their appearance in **Table 1**.

To demonstrate the locality of long- and short-range perturbations, we introduce a single perturbation at one chosen location in the structure and measure the mean absolute error of the onsite Hamiltonian blocks when compared to that of the unperturbed structure. The types of perturbations introduced include translation of a hafnium atom, oxygen vacancy, and substitution of an oxygen atom with a hafnium one, and are plotted against the distance from perturbation in **Fig. 10** (a), (b), and (c), respectively.

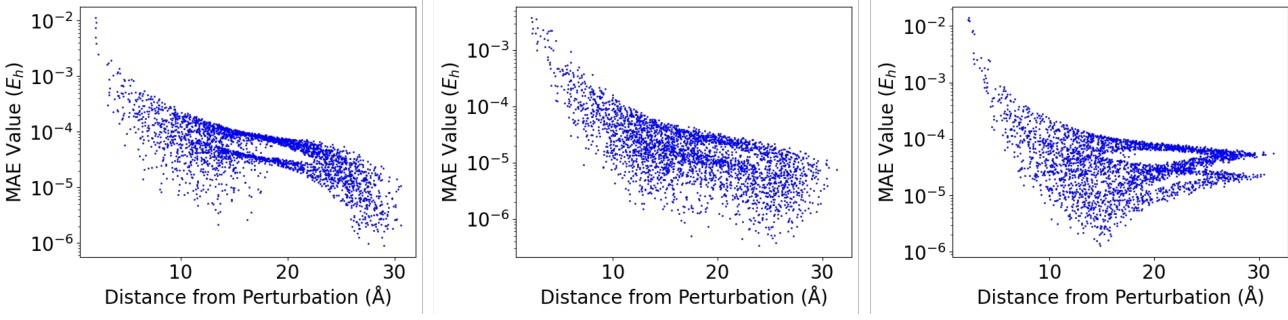

*Figure 10.* Scatter plots showing the decay of MAE with increasing distance from different perturbations, including (a) 0.1 Å translation of a Hf atom, (b) replacement of O atom with a vacancy, and (c) replacement of O atom with Hf atom.

In all cases, the effect of the perturbation rapidly decays with increasing distance. For the case of the 0.1 Å translation, the average onsite MAE at a distance of 8 Å away is given by $0.15 \ mE_h$. Considering the average value of an onsite Hamiltonian block ($63 \ mE_h$), the perturbation only affects the matrix elements by 0.24% overall. Similarly, for vacancy and substitution perturbations, the matrix elements of atoms located 8 Å away only changed by 0.18% and 0.12% respectively.

This implies that for our chosen cutoff of 8 Å, perturbations occurring outside of the radius surrounding the atom have a negligible effect on its Hamiltonian matrix elements. This also indicates that the electronic structure of that atom can be learned using information from the local atomic environment.

## G. Cutoff radius and connectivity

We now explore the minimal graph connectivity that can be used by the network to accurately learn relevant features of $HfO_2$. To do this we use the slice partition approach introduced in **Section 3.3**, using 18 slices of length $t_{slice} = 3$ Å to train the network. Results are reported in (**Table. 8**). Reducing the value of $r_{cut}$ below 8 Å noticeably increases the error ($\epsilon_{n/e}$), thus demonstrating the sensitivity of $H$ to the exact connectivity of the graph. Going from $r_{cut} = 8$ Å to 12 Å, the edge error improves, while the node error begins to plateau, but the node degree (which is proportional to the memory consumption of the network) grows by $1.7\times$. This is consistent with the negligible changes to the eigenvalue spectra with $r_{cut}$ of 8 Å (**Fig. 9**).

| $r_{cut}$ [Å] | $deg(n)$ | $deg(n)'$ | $\epsilon_{\mathbf{n}}[mE_h]$ | $\epsilon_{\mathbf{e}}[mE_h]$ | $\epsilon_{\mathbf{tot}}[mE_h]$ | $\epsilon_{\mathbf{tot}}[meV]$ |
|---|---|---|---|---|---|---|
| 4 | 21.02 | 10.60 | 2.84 | 0.52 | 0.64 | 17.50 |
| 6 | 72.41 | 32.01 | 2.27 | 0.26 | 0.29 | 7.98 |
| 8 | 177.61 | 49.89 | 2.16 | 0.16 | 0.18 | 4.84 |
| 10 | 347.16 | 78.85 | 2.18 | 0.12 | 0.12 | 3.38 |
| 12 | 590.25 | 138.59 | 2.13 | 0.09 | 0.10 | 2.58 |

*Table 8.* Prediction accuracy of the network with different $r_{cut}$. Training was done with a single slice of length $t_{slice} = 3$ Å. The edge connectivity of the matrix is set by $r_{cut}$. $deg(n)$ is the average node degree, and $deg(n)'$ the reduced node degree omitting virtual node neighbors. Note that for this value of $t_{slice}$, the majority of neighbors for the average node are virtual. $\epsilon_n$ and $\epsilon_e$ are the Mean Average Error (MAE) for nodes/edges, respectively. All units are in $[\times 10^{-3} E_h]$. The validation loss of the model is computed from a slice of similar length extracted from structure 1. The networks are tested on an unseen full graph (structure 3) constructed with the same $r_{cut}$.

We perform a similar study on the cutoff radius of PtGe in Table 9. Increasing the cutoff radius once again increases the overall prediction accuracy of the trained model, mostly due to improvements in edge-prediction accuracy.

| Cutoff [Å] | $deg(n)$ | $deg(n)'$ | $\epsilon_{\mathbf{n}}[mE_h]$ | $\epsilon_{\mathbf{e}}[mE_h]$ | $\epsilon_{\mathbf{tot}}[mE_h]$ | $\epsilon_{\mathbf{tot}}[meV]$ |
|---|---|---|---|---|---|---|
| 4 | 14.28 | 10.00 | 1.08 | 0.26 | 0.33 | 8.96 |
| 6 | 50.17 | 27.37 | 0.80 | 0.14 | 0.16 | 4.28 |
| 8 | 118.71 | 51.57 | 0.77 | 0.08 | 0.09 | 2.49 |
| 10 | 234.28 | 84.07 | 0.80 | 0.08 | 0.08 | 2.17 |

*Table 9.* Prediction accuracy of model on amorphous PtGe material with different $r_{cut}$

## H. Compute environment and runtime comparisons

The training is performed with PyTorch Distributed Data Parallel (Li et al., 2020), where the graph partitions (slices) can be distributed between GPUs.

### H.1. Memory consumption of full-graph training

During the training of the full graph model, the peak memory consumption observed was 61.68 GiB on a single NVIDIA A100 GPU. Most of the consumption does not stem from the network and the structure but from the additional memory needed for the convolution operations.

### H.2. Scalability of augmented partitioning

In **Fig. 11**, we show the decrease in time per epoch and resulting speedup when using the *augmented partitioning* approach introduced in **Section 3.3**. Since the partitions are independent, the only communication involved in every epoch is a collective to inform each GPU/rank of the loss of each other rank. The time per epoch thus decreases uniformly with the number of slices ($N_t$) used.

Despite the independence of each batch and the minimal communication per epoch, the scaling is not perfectly linear. The deviation from an ideal speedup can be attributed to two factors:

- Load imbalance: The partitioning approach was designed to leverage the periodicity in the $y$- and $z$- direction within a straightforward implementation. However, it is not ideal in terms of the number of cuts/number of virtual nodes/edges required, resulting in a slightly different amount of work per rank which leads to an observable load imbalance at higher $N_t$. This effect can be seen in the allocated memory per partition (**Fig. 11(c)**). We note that the *augmented partitioning* method can be used with any standard graph-partitioning algorithm.

- Computational overhead of the virtual nodes and edges: Individual nodes and edges of the graph can be repeated in labeled and virtual node lists. Treating the replicas introduces additional computational cost while training the network, which increases with $N_t$. This overhead is maximum with the use of very small slices (large $N_t$), thus introducing a trade-off between parallelism and time per epoch.

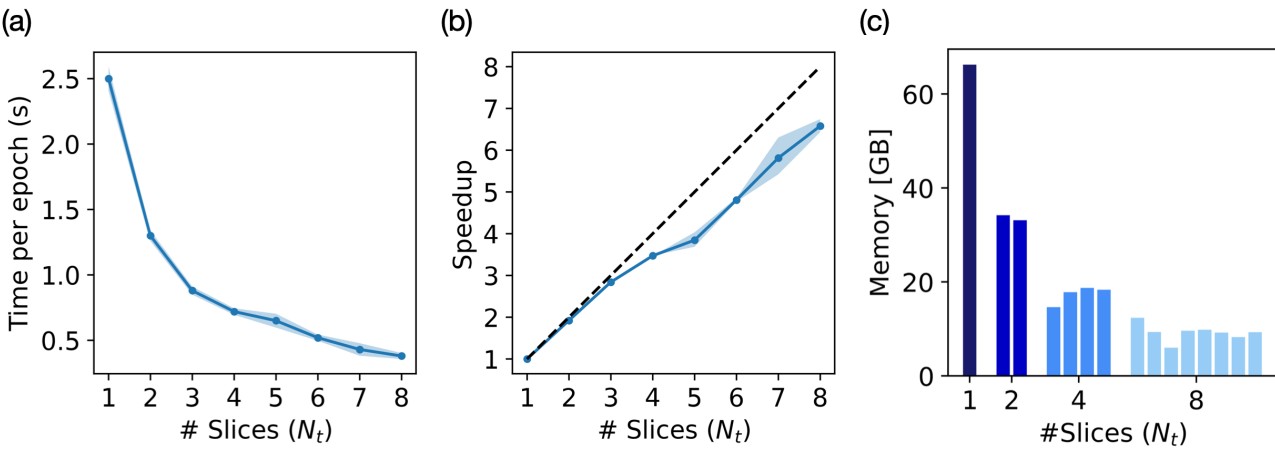

*Figure 11.* (a) Time per epoch and (b) speedup resulting from the use of increasing numbers of slices $N_t$. Median values are shown, while the error bands are one standard deviation. Experiments were run on NVIDIA A100 GPUs with # ranks set to $N_t$. Measurements are only shown up to 8 slices/8 GPUs due to limitations in available compute resources at the time of submission. The fill-between indicates the range in runtime over the first 30 minutes of training. The dashed black line corresponds to the ideal speedup, in which case the use of $N_t$ slices would enable an $N_t\times$ speedup in the runtime per epoch. (c) Measured peak memory consumption as a function of the number of partitions, where each bar corresponds to a different GPU. Variation in memory consumption between GPUs at each individual value of $N_t$ translates to load imbalance, which correlates with the deviation from ideal scaling shown in (b).

### H.3. $H_2O$ vs $HfO_2$ runtimes

In **Section 4.5**, we make a comparison between the computational cost of computing the Hamiltonian for an $H_2O$ molecule and the $HfO_2$ structure. To approximate the cost of generating them under the same computational conditions, we set up CP2K simulations with a DZVP basis for $H_2O$. The computation time per $H_2O$ molecule was 7s, when run on 12 nodes with 12-core Intel Xeon E5-2680 CPUs and NVIDIA P100 GPU, resulting in a total of 0.04 node hours. The $HfO_2$ structures require 3.65 node hours in the same compute environment (but distributed to 27 nodes). The difference, omitting scaling behavior, is roughly $\sim 100\times$.

## I. Additional Tests

### I.1. Sub-stoichiometric hafnium oxide

Here, we provide a more detailed study on the prediction of substoichiometric $HfO_2$, where the train/test structures contain different vacancy fractions. 18 slices ($t_{slice} = 3$ Å) were used in all cases. The results are summarized in Table 10. The $\epsilon_n$ and $\epsilon_e$ values across different experiments lie within a small range (2.24-2.73 $mE_h$ and 0.13-10.18 $mE_h$ respectively), showing that the network generalizes well to structures of different vacancies, regardless of which vacancy configuration it was trained

| Stoichiometry ($x$) | | $\epsilon_\mathbf{n}\,[mE_h]$ | $\epsilon_\mathbf{e}\,[mE_h]$ |
|---|---|---|---|
| Training | Testing | | |
| 1.9 | 1.9 | 2.24 | 0.13 |
| 1.9 | 1.8 | 2.44 | 0.14 |
| 1.9 | 1.7 | 2.73 | 0.15 |
| 1.8 | 1.9 | 2.34 | 0.15 |
| 1.8 | 1.8 | 2.38 | 0.15 |
| 1.8 | 1.7 | 2.45 | 0.16 |
| 1.7 | 1.9 | 2.32 | 0.14 |
| 1.7 | 1.8 | 2.37 | 0.15 |
| 1.7 | 1.7 | 2.51 | 0.18 |

*Table 10.* $HfO_x$ models trained and tested with different stoichiometry ($x$) using augmented partitioning. 18 slices of each structure, each 3 Å thick, was used for training. The training method is identical to the one used to obtain Table 3. Models trained on $HfO_{x=1.9}$ (5%), $HfO_{x=1.8}$ (10 %), and $HfO_{x=1.7}$ (15% vacancies) are tested on test structures with vacancies ranging from 5% to 15%.

on. To demonstrate that the *augmented partitioning* approach similarly does not affect accuracy for sub-stoichiometric $HfO_x$, we also perform full graph training using structure 2 with 15% vacancies, and compare with the *augmented partitioning* approach in Table 12. The comparison of $\epsilon_n$ and $\epsilon_e$ values between full and partitioned approaches indicates that both approaches generalize well to different stoichiometry. These values are also close to that of stoichiometric $HfO_2$ in Table 3.

### I.2. Partitioning tests for other materials

For each material, we train the model on partitions of different thicknesses containing the same number of atoms. The results are summarized in Table 11. The same test was also repeated on substoichiometric $HfO_{1.7}$ and tested on structures with different stoichiometry (shown in Table 12. It can be seen that augmented partitioning preserves the accuracy of prediction for all material cases studied. For $HfO_x$, it also preserves the model's ability to generalize to structures of different stoichiometry.

| Material | $t_{slice}$[Å] | $N_t$ | $r_{cut}$[Å] | $N_n$ | $N_e$ | $\epsilon_\mathbf{n}[mE_h]$ | $\epsilon_\mathbf{e}[mE_h]$ | $\epsilon_\mathbf{tot}[mE_h]$ | $\epsilon_\mathbf{tot}[meV]$ |
|---|---|---|---|---|---|---|---|---|---|
| a-GST | 30 | 1 | 12 | 710 | 138532 | 1.15 | 0.08 | 0.08 | 2.28 |
| a-GST | 5 | 6 | 12 | 710 | 48356 | 0.97 | 0.08 | 0.09 | 2.40 |
| a-PtGe | 50 | 1 | 8 | 1633 | 182372 | 0.77 | 0.08 | 0.08 | 2.26 |
| a-PtGe | 5 | 10 | 8 | 1633 | 84416 | 0.77 | 0.08 | 0.09 | 2.49 |

*Table 11.* Summary of model performance when trained on amorphous GST and PtGe structure with slices of different thicknesses. The structures used for training, validation, and testing are displayed in Table 1. $N_n$ and $N_t$ refers to the total number of labeled nodes and edges in the training set respectively.

| Training method | Stoichiometry ($x$) | $\epsilon_\mathbf{n}\,[mE_h]$ | $\epsilon_\mathbf{e}\,[mE_h]$ |
|---|---|---|---|
| | Testing set | | |
| partitioned | 1.9 | 2.32 | 0.14 |
| partitioned | 1.8 | 2.37 | 0.15 |
| partitioned | 1.7 | 2.51 | 0.18 |
| full | 1.9 | 3.02 | 0.12 |
| full | 1.8 | 2.54 | 0.11 |
| full | 1.7 | 2.52 | 0.11 |

*Table 12.* Comparison between full graph training and the augmented partitioning training using the same $HfO_{1.7}$ structure with 15% vacancies. Models are tested on structures with vacancies ranging from 5% to 15%.

### I.3. Small Molecule Benchmarks

We test our backbone architecture setup on part of the MD17 benchmark (Schütt et al., 2019), which consists of small-molecules, each with 3 (water), 9 (malondialdehyde) or 12 (uracil) atoms. The hyperparameters used are listed in **Table. 14**.

| Dataset | Model | Train | Validate | Test | Batch size | $\epsilon_{\mathbf{tot}}\,[\mu E_h]$ |
|---|---|---|---|---|---|---|
| water | QHNet | 500 | 500 | 3900 | 10 | 10.79 |
|  | This work | 500 | 500 | 3900 | 10 | 5.60 |
| malondialdehyde | QHNet | 25,000 | 500 | 1,478 | 5 | 21.52 |
|  | This work | 25,000 | 50 | 1,478 | 5 | 16.60 |
| uracil | QHNet | 25,000 | 500 | 4,500 | 5 | 20.12 |
|  | This work | 25,000 | 50 | 4,500 | 5 | 13.80 |

*Table 13.* Comparison with QHNet on MD17 benchmark dataset containing water, malondialdehyde and uracil

The number of training steps for each experiment is fixed at 200,000 to match that of QHNet in (Yu et al., 2023b) . The results in **Table 13** show that the performance of our setup is comparable to that of the baseline for all measured cases.

| Hyper-parameters | MD 17 dataset |
|---|---|
| Optimizer | Adam |
| Precision | double (f64) |
| Scheduler | ReduceLROnPlateau |
| Criterion | RMSE + MAE |
| Initial learning rate | $1 \times 10^{-3}$ |
| Minimum learning rate | $1 \times 10^{-10}$ |
| Number of MP Layers | 2 |
| Decay patience $t_{patience}$ | 50 |
| Decay factor | 0.5 |
| Threshold | $1 \times 10^{-5}$ |
| Maximum degree $L_{max}$ | 4 |
| Maximum order $M_{max}$ | 4 |
| Embedding size | 128 |
| Number of attention heads $N_h$ | 2 |
| Feedforward Network Dimension | 64 |

*Table 14.* Hyper-parameters used for MD17 dataset.

