# OpenReview forum: "Learning the Electronic Hamiltonian of Large Atomic Structures"
_ICML.cc/2025/Conference — ICML 2025 poster_

### Official Review · Reviewer_nuFS · 2025-02-25

**Overall Recommendation:** 2

**Summary:**

This work focuses on scaling Hamiltonian prediction to large periodic structures. While a common ML problem, Hamiltonian prediction has been challenging in large structures due to the quadratic scaling of the Hamiltonian. The authors propose a partitioning scheme to enable distributed computing and batching of large unit cells. Further, the authors present a new dataset consisting of 3 periodic large compounds with a total of 7 structures. Due to the size of the structures, training on a single sample suffices.

## After the rebuttal

I appreciate the authors’ response but remain with my many points of my initial critisim. Primarily, comparisons to previous works are possible by evaluating on smaller datasets. While new datasets are always welcome this does not rule out the evaluation on preivous established ones. This undermines their second contribution as listed by the authors. Overall, I do not strongly object an acceptance but also do not recommend it.

**Claims And Evidence:**

The following claims are made:
1. A single message passing GNN achieves accurate Hamiltonian prediction on large structures.
2. The authors demonstrate that Hamiltonian prediction scales to thousands of atoms.
3. The proposed partitioning scheme enables efficient training on large compounds.

In general, these claims also seem supported but the limited evaluation makes the specific judgements hard.

**Essential References Not Discussed:**

-

**Experimental Designs Or Analyses:**

The authors evaluate only on their newly proposed dataset their newly propose method. This combination poses a challenge for the reader to identity meaningful transferable improvements. Evaluation on standard datasets for their proposed GNN and previous GNNs on their new dataset would strengthen the work significantly and would improve its contextualization. Additional downstream application, e.g., by running DFT, or computing observables would strengthen the paper's position.

**Methods And Evaluation Criteria:**

The author propose a new GNN for predicting Hamiltonians that is similar to previous works but only uses a single message passing step. They evaluate it with their partitioning scheme.

**Other Comments Or Suggestions:**

* Please use consistent units, using eV and Eh in different places makes comparisons more difficult.
* l. 338 should be Table 3.

**Other Strengths And Weaknesses:**

Strengths:
* The paper is well written and easy to understand if one has the necessary background.
* The problem is interesting and the results are encouraging for large-scale Hamiltonian prediction.

Weaknesses:
* The works presents limited theoretical and technical contributions and heavily relies on its empirical results.
* The empirical evaluation is quite limited and does not allow the reader to identify the downstream performance or comparisons to previous works.

**Questions For Authors:**

I am willing to increase my score if the authors manage to clearly isolate their contributions and show the evaluate in each contribution via ablation studies, comparisons with previous works, and evaluation on established datasets.

**Relation To Broader Scientific Literature:**

The authors should consider contextualizing their work with the batching works done in graph neural networks where similar issues for large graphs arise.

**Theoretical Claims:**

The paper does not introduce theoretical claims.

---

> ### Author Rebuttal · Authors · 2025-03-28
>
> **Isolation of Contributions:** Our two main contributions not found in other works are:
>
> 1. An augmented partitioning approach that allows arbitrarily large graphs to be broken down into independent partitions that (1) maintain the connectivity of the full structure through virtual nodes/edges, (2) fit into GPU memory during training, and (3) ensure high scalability without compromising achievable test accuracy.
>
> 2. A GNN model that is custom-made for scalable Hamiltonian predictions of large, complex atomic structures. Unlike previous models [3-8], it combines strict locality with efficient SO(2) convolutions plus multi-head equivariant attention to better distinguish between complex atomic environments.
>
> **Ablation studies:** An ablation study of our proposed augmented partitioning approach is shown in **Table 2 of the results section**, where the omission of virtual components led to a 50% increase in node error and an 88% increase in edge error, as the graph tries to aggregate information through an incomplete graph structure. The strict locality is also motivated and justified by previous work like Allegro [5,8], plus we have also conducted studies on their effects on Hamiltonian matrices in **Appendix F**.
>
> **Discussion of other batching methods for large graphs:** Previous popular works on large graph partitioning [9] employ neighborhood sampling approaches to reduce communication volume between partitions. This cannot be applied to Hamiltonian predictions, as omitting any graph connections leads to the wrong atomic neighborhood. Our method is a novel way to (1) batch the data, and (2) removes the need for communication between different partitions while maintaining graph connectivity.
>
> **Evaluation on established datasets:** Established Hamiltonian datasets (e.g., QH9 [6]) consist of small molecules, which cannot be used to evaluate a model's performance on the large-scale problem we are trying to tackle. The use of custom datasets is also encouraged in the applications track, where ML is applied to a wide variety of problems that have yet to be tackled, and a valid comparison does not yet exist.
>
> **For evaluating previous methods on our custom dataset, the closest accuracy comparison (within memory limits) is shown in Table 3 in the paper for HfO2 (and also in Table 11 in Appendix I.3 for GST and PtGe)**. Full graph training is the default baseline approach used by previous SOTA. We showed that for all three datasets, the application of augmented partitioning approach maintains accuracy, while significantly increasing scalability (shown in Appendix H.2). The values we obtained (0.99-5.16 meV) are also within range of what SOTA models obtained (1.5-3.23 meV) for simpler structures [3-4].
>
> **Downstream applications:** We applied our approach to the simulation of valence change memory **in the response to Reviewer Kuqs**. The obtained accuracy allows us to observe trends in resistance contrast changes as a function of applied voltage and resulting ion movements. By subverting expensive DFT calculations entirely, we enable a range of previously unfeasible large-scale device simulations. **The true meaning of our work, therefore, lies in its ability to address a crucial bottleneck that previous methods (including DFT) could not.** It also sets a precedent for the prediction of complex structures at the scale of ~10^3 atoms per unit cell and can be used as an initial benchmark for future models that also aim to do so for various applications.
>
> **References**
>
> [1] Jónsson, H., Mills, G. & Jacobsen, K. W. in Classical and Quantum Dynamics in Condensed Phase Simulations (eds Berne, B. J. et al.) 385–404 (World Scientific, 1998)
>
> [2] Kaniselvan, M., Luisier, M., and Mladenovic, M. An atomistic model of field-induced resistive switching in valence change memory. ACS Nano, March 2023.
>
> [3] Wang, Y., Li, Y., et. al. Universal materials model of deep-learning density functional theory hamiltonian. Science Bulletin, 69(16):2514–2521, 2024b.
>
> [4] Zhong, Y., Yu, H., Su, M., Gong, X., and Xiang, H. Transferable equivariant graph neural networks for the hamiltonians of molecules and solids. npj Computational Materials, 2023.
>
> [5] Musaelian, A., Batzner, S., Johansson, A., Sun, L., Owen, C. J., Kornbluth, M., and Kozinsky, B. Learning local equivariant representations for large-scale atomistic dynamics. Nature Communications, 14(1):579, 2023
>
> [6] Yu, H., Liu, M., Luo, Y., Strasser, A., Qian, X., Qian, X., and Ji, S. Qh9: A quantum hamiltonian prediction benchmark for qm9 molecules, 2023a.
>
> [7] Y Li, et. al Enhancing the scalability and applicability of Kohn-Sham Hamiltonian for scalable molecular systems. ICLR 2025.
>
> [8] Zhouyin, Z., et. al. Learning local equivariant representations for quantum operators, ICLR 2025
>
> [9] M. Besta and T. Hoefler, "Parallel and Distributed Graph Neural Networks: An In-Depth Concurrency Analysis," in IEEE Transactions on Pattern Analysis and Machine Intelligence

---

### Official Review · Reviewer_zXDf · 2025-03-01

**Overall Recommendation:** 3

**Summary:**

The author propose a graph partitioning strategies to localize message passing and new networks which leverages SO(2) convolution for predicting the Hamiltonian for large atomic structures. The author demonstrates its capabilities by predicting the electronic Hamiltonian of various systems with up to 3,000 nodes with and ≤0.55% error in the eigenvalue spectra.

**Claims And Evidence:**

Yes

**Essential References Not Discussed:**

Some other paper also leverages the SO(2) convolution such as [1].

[1] Enhancing the scalability and applicability of Kohn-Sham Hamiltonian for scalable molecular systems. ICLR 2025.

**Experimental Designs Or Analyses:**

The derived Hamiltonian should also be evaluated with SCF iterations speedup or the MAE on system energy deriving from the Hamiltonian.

**Methods And Evaluation Criteria:**

Yes

**Other Comments Or Suggestions:**

N/A

**Other Strengths And Weaknesses:**

1. Strength: The significance of this work is high, it could be a promising avenue for large-scale simulation or all-atom protein simulations.
2. Weakness: The evaluation on the real-world applications is very limited (such as SCF speedups, system energy prediction, dipole moment prediction). The graph partition strategy is not that novel, and the author did not addresses the non-diagonal blocks beyond cutoffs. Missing of the long-range interactions could potentially impact the real-world applications. The graph-partitioning could also potentially lead to non-smoothness in real world simulations.

**Questions For Authors:**

1. How does the network handle the non-diagonal part of the Hamiltonian beyond cutoffs? Methods like QHNet leverages the pair construction layer which performs a tensor product between every pair of the atoms. I think it's not doable here.
2. What is the absolute error for the eigen-spectrum, is it within the chemical accuracy? (< 1 kcal/mol)
3. The graph would generate multiple graph slices, how do you combine them to generate the final H (a diagram or algorithm would suffice)  ? And for the embedding of virtual nodes, will their embedding be updated by the embedding from other graph slices the next time you encounter the graph?

**Relation To Broader Scientific Literature:**

It is somewhat relavent, it could be broadly applicable to distributed graph computing.

**Theoretical Claims:**

N/A

---

> ### Author Rebuttal · Authors · 2025-03-28
>
> **Non-diagonal part of Hamiltonian beyond cutoffs:**
>
> **The use of a cut-off radius is standard practice in Hamiltonian prediction literature of bulk structures [3-4], which exploits the nearsightedness of the Hamiltonian matrix.** **Appendix F** include two studies on the nearsightedness of Hamiltonians for our dataset:
>
> 1. **In Fig. 7**, we illustrated the decay in values of Hamiltonian elements with increasing inter-atomic distance for all three datasets.
>
> 2. **In Fig. 8** we quantify the effect of discarding orbital blocks beyond a predefined cutoff on the prediction accuracy. Removing those with rcut > 4 Angstroms (the chosen rcut for HfO2 in the study) from the DFT reference data showed a negligible difference in the eigenvalue spectra of H.
>
> **Long range interactions beyond cutoff:**  In **Appendix F, Fig. 9** we studied the effects of various types of perturbations (shift, vacancy, substitution) beyond the cutoff radius and showed that they all have a negligible effect on the Hamiltonian matrix elements (less than 0.24%). This indicates that the Hamiltonian of the atoms in our datasets can be learnt from their local atomic environment. The use of strict locality is also well-established in literature, and previous architectures (e.g., Allegro for force fields [5] and [8]) have used this to improve scalability while still achieving state-of-the-art prediction accuracy.
>
> **Absolute Error for Eigenspectrum:** The threshold of < 1 kcal/mol is normally used for the prediction of thermochemical quantities (e.g., interaction, ionization energy) and is not commonly applied to eigenspectra of bulk structures with several atoms. Still, we understand the need for a quantitative measure with units, so we computed the mean absolute error for all eigenvalues, obtaining 3.08 mHartrees (2.53 mHartrees for occupied energies).
>
> **Real World Applications:** Unlike the molecular Hamiltonians predicted by models like WANet, whose applications include the extraction of molecular quantities (e.g., orbital energies), the Hamiltonians of large structures (> 10^3 atoms) targeted by our approach have their own set of downstream applications (e.g., semiconductor physics). To truly assess its practical relevance in this domain, we applied it to the simulation of valence change memory cells, **with details outlined in the response to Reviewer Kuqs**. Using predicted Hamiltonians for simulations, we achieved levels of accuracy that allow us to observe key trends in current vs. voltage characteristics of VCM cells and reveal the dependence of the current on the atomic geometry of these devices. In general, it enables large-scale simulation workflows involving repeated DFT updates over 100-1000 time steps that were previously unfeasible, and also allows large structures beyond DFT capabilities (10k+ atoms) to be simulated.
>
> **Multiple graph slices and Hamiltonian construction:** In this paper, slices are only used to train the model. The final trained model was then used to predict the full structure/graph at once (e.g., all 3000 atoms), with only one final predicted Hamiltonian matrix obtained, so there is no need to assemble any slices. For very large structures, the Hamiltonian can also be predicted separately as different slices with multiple GPUs/CPUs. Since the nodes and edges in each slice define an independent sub-block, the final matrix can then be reconstructed by populating it piece by piece with the components of each partition. Both approaches lead to the same result.
>
> **Role of virtual nodes/edges (whether they are updated by other graph slices):** We do not update the embeddings (outputs) of the virtual nodes and edges in any case. In the paper, we mentioned that their output embeddings have no meaning/do not exist (hence the term virtual), and are not involved in training and predictions. Their only purpose lies in their initialized inputs (atomic numbers and distances), which are used to inform the labelled (non-virtual) atoms within the partition of their correct one-hop atomic neighborhood. The use of only the inputs of virtual nodes and edges to maintain connectivity without communication is one of the key novel contributions of our approach.
>
> **Note that the list of references is found in the response to Reviewer 4 (nuFS)**

---

> > ### Comment · Reviewer_zXDf · 2025-04-06
> >
> > I thank the authors for their responses. I am willing to increase my score to 3.

---

### Official Review · Reviewer_9oL5 · 2025-03-08

**Overall Recommendation:** 4

**Summary:**

The paper proposes a new method for applying GNNs to learn the electronic Hamiltonian for atomic structures beyond the unit cell. The paper starts by introducing the use of GNNs for property prediction on fairly small size unit cell materials and molecules and motivates the need to capture more complex materials behaviors, such as defects and strains, at larger size scales. To enable GNNs to be applied for larger scale atomic systems, the paper proposes two contributions: 1. Learning the ground-state Hamiltonian using a SO(2)-equivariant GNN that learns local embeddings for the diagonal and off-diagonal blocks of the Hamiltonian. 2. A graphs partitioning method that enables partitioning larger input graphs, which represent larger atomic systems, to more effectively deploy the proposed GNN. The final paragraph of the introduction highlights that the authors deploy their method on amorphous systems, which include various defect types thereby providing a reasonable test case.

Section 2 describes relevant background for electronic structure modeling related to energy levels and wavefunctions that make up the Hamiltonian matrix. The Hamiltonian matrix is computed based on a basis of atomic orbitals that transform under spherical harmonics. The Hamiltonian matrix itself can be decomposed into sub-matrices that describe the interactions of orbitals on the same atom (diagonals) and between different atoms (off-diagonal). The main challenge with calculating Hamiltonian matrices is the computational cost, which has lead to approximations that rely on creating a repeatable unit cell of atoms. That unit cell, however, often fails to properly capture atomic behavior that require larger size, such as defects. Section 2 also describes related work on using machine learning for directly predicting the Hamiltonian matrix, focusing mostly on equivariant GNNs that operate in a spherical harmonic basis.

Section 3 introduces the primary method of the paper starting with the representation for the block of the Hamiltonian matrix and the proposed network architecture. The network architecture mainly relies on a combination of eSCN equivariant convolution to learn over the atomic graph and multi-headed attention to learn in embeddings for the local atomic environments. Section 3.3 describes the graph partitioning method that enables the proposed method to model interactions beyond local atomic interactions while still maintaining a local receptive field for scalability. The proposed partitioning approach leverages virtual nodes and edges that augment the graph for a given partition but are not used for the message passing. As such, the receptive field includes only a 1-hop neighborhood ensuring scalability to larger systems while maintaining sufficient accuracy. Section 3.4 describes the dataset generation focusing on moelcular dynamics trajectories of amorphous materials, namely: HfO2, GeSbTe, and PtGe. The systems span thousands of atoms with >10k orbitals and >100k edges.

Section 4 presents the main results. The first set of experiments presents ablations related to the proposed graph partition mechanism and generally find that partitioned graphs achieve competitive results when compared to full-graph training. The results in Section 4.2 claim that the proposed method can effectively capture the structural disorder contained in the amorphous materials data. Section 4.3 claims that the proposed method can capture compositional disorder in HfO2 and Section 4.4 outlines a case study comparing predictions of the full Hamiltonian using the trained network and various partition slices. Section 4.5 describes computational savings achieved by the graph partitioning approach outlining both compute and memory savings. Section 5 provides a brief conclusion.

## update after rebuttal

The authors answered my questions to my satisfaction and agreed to include relevant details that will make the paper stronger, including clarifying related work for their paper and providing a comparison on scalability. As such, I increased my score and support of the paper.

**Claims And Evidence:**

The primary claims are backed up with evidence from targeted experiments on large-scale atomic systems. The method descriptions are through and provide relevant details for both the physics and machine learning considerations.

**Essential References Not Discussed:**

I encourage the authors to discuss the references in the above box. The papers related to challenges for GNNs in simulations appear particularly relevant. The paper could also benefit by providing more details on how their work exceeds the capabilities of prior work, such as Allegro [1] and Allegro-Legato [2] which is mentioned in the paper, that have tried scaling GNNs to large scale simulation.

The paper should also discuss relevant context for Hamiltonian Neural Networks [3] as a general method.

[1] Musaelian, A., Batzner, S., Johansson, A., Sun, L., Owen, C.J., Kornbluth, M. and Kozinsky, B., 2023. Learning local equivariant representations for large-scale atomistic dynamics. Nature Communications, 14(1), p.579.

[2] Ibayashi, H., Razakh, T.M., Yang, L., Linker, T., Olguin, M., Hattori, S., Luo, Y., Kalia, R.K., Nakano, A., Nomura, K.I. and Vashishta, P., 2023, May. Allegro-legato: Scalable, fast, and robust neural-network quantum molecular dynamics via sharpness-aware minimization. In International Conference on High Performance Computing (pp. 223-239). Cham: Springer Nature Switzerland.

[3] Greydanus, S., Dzamba, M. and Yosinski, J., 2019. Hamiltonian neural networks. Advances in neural information processing systems, 32.

The paper would also benefit from discussing prior work on graph partitioning for GNNs [4] in the context of the implemented partition algorithm. The algorithm in [4] seems to have a similar approach and claimed benefits.

[4] Mostafa, H., 2022. Sequential aggregation and rematerialization: Distributed full-batch training of graph neural networks on large graphs. Proceedings of Machine Learning and Systems, 4, pp.265-275.

**Experimental Designs Or Analyses:**

The experiment design for the three amorphous materials, described in the main paper and supplementary material, are generally sound and detailed.

**Methods And Evaluation Criteria:**

The methods and evaluation criteria used are created by the authors themselves and are generally well chosen to demonstrate their proposed method. The experiments could be strengthened by describing why relevant baselines (e.g. Allegro) may not be able to scale to the systems used.

The paper could be strengthened by showing additional experiments on smaller systems - ideally these results would show that minimal performance drops occur with smaller (unit cell scale) systems, providing more evidence for the capabilities of the proposed method.

**Other Comments Or Suggestions:**

N/A

**Other Strengths And Weaknesses:**

Overall, the paper presents an interesting and potentially compelling method for scaling GNNs to larger materials systems. The paper could be strengthened by:

* Providing more details on whether the models evaluated are one for each system or a generalized model.
* Whether their proposed method would be applicable to smaller systems without performance drops. As mentioned before, this would help strengthen the reasons for using the method.
* Discussing whether the graph partitioning approach can be applied to any model architecture and what limitations may exist.

**Questions For Authors:**

1. Did you train one model for each material system?
2. Do you think your approach can generally across different structures and systems? What would be needed for that?
3. Can you clarify if there is a different in the cut-off distance between slices and the cut-off distance used for graph creation?

**Relation To Broader Scientific Literature:**

The paper generally covers a good amount of the relevant literature. It could be further strengthened by discussing references related to the challenges of applying GNNs directly in MD simulation [1][2][3], providing broader context for GNN-based methods for atomistic modeling [4] and tying the challenges to broader applications [5].

[1] Fu, X., Wu, Z., Wang, W., Xie, T., Keten, S., Gomez-Bombarelli, R. and Jaakkola, T., Forces are not Enough: Benchmark and Critical Evaluation for Machine Learning Force Fields with Molecular Simulations. Transactions on Machine Learning Research.

[2] Bihani, V., Mannan, S., Pratiush, U., Du, T., Chen, Z., Miret, S., Micoulaut, M., Smedskjaer, M.M., Ranu, S. and Krishnan, N.A., 2024. EGraFFBench: evaluation of equivariant graph neural network force fields for atomistic simulations. Digital Discovery, 3(4), pp.759-768.

[3] Gonzales, C., Fuemmeler, E., Tadmor, E.B., Martiniani, S. and Miret, S., 2024. Benchmarking of Universal Machine Learning Interatomic Potentials for Structural Relaxation. In AI for Accelerated Materials Design-NeurIPS 2024.

[4] Duval, A., Mathis, S.V., Joshi, C.K., Schmidt, V., Miret, S., Malliaros, F.D., Cohen, T., Liò, P., Bengio, Y. and Bronstein, M., 2023. A hitchhiker's guide to geometric gnns for 3d atomic systems. arXiv preprint arXiv:2312.07511.

[5] Miret, S., Lee, K.L.K., Gonzales, C., Mannan, S. and Krishnan, N.M., 2025. Energy & Force Regression on DFT Trajectories is Not Enough for Universal Machine Learning Interatomic Potentials. arXiv preprint arXiv:2502.03660.

**Theoretical Claims:**

The theoretical claims presented are generally well supported.

---

> ### Author Rebuttal · Authors · 2025-03-28
>
> **1. One model for each material system**: Yes, the models are trained for a specific material system, similar to most other literature ([4-8]). The current work does not demonstrate a universal Hamiltonian prediction model, though this can be achieved with a larger library of datasets, as shown by [3]. No public database for structures with 1000+ atoms per unit cell currently exists, so the necessary data must be manually generated. Data creation is computationally expensive due to the melt-and-quench and relaxation processes required. With enough collective effort and time, however, a universal model for all materials, including large, amorphous structures, can be feasibly trained with our approach.
>
> **2. Generalization to different structures and systems**: Our approach (strictly local model + augmented partitioning) can be straightforwardly applied to other materials systems, including periodic structures (crystals, 2D materials) and molecules. We also applied it to the prediction of crystalline HfO2 structure and obtained far more accurate predictions (1.1 meV MAE) relative to amorphous HfO2 (5.16 meV) without any further optimization. Molecular examples are also included in the code attached in supplementary materials (though augmented partitioning and strict locality are excessive in these contexts due to their small, isolated systems and information-poor atomic neighborhoods). Here, the small number of atoms per unit cell, plus the larger similarity between training and test datasets, makes higher accuracies much more achievable.
>
> **3. Cutoff for graph and cutoff for slice**: The cutoff (rcut) used to define the graph connectivity is not related to the slice length (t) used to define the partitions. They can be independently chosen.
>
> **Applicability of partitioning to other architectures**: Augmented partitioning can be applied to other strictly local architectures (e.g., Allegro [5]), since it is implemented during graph construction.**
>
> **How our work exceeds previous work:**
>
> Firstly, for Allegro, the prediction task is for force and energies. The computational challenge they encounter is fundamentally different from ours, and comparisons are not possible as:
>
> (1) Energy and force predictions involve only node embeddings, while Hamiltonian predictions also include edge embeddings to capture off diagonal blocks, and edges far outnumber the nodes (e.g. 3000 atoms, 500,000+ edges)
>
> (2) H predictions require higher degree tensors to capture all orbital interactions.
>
>
> **Scalability comparisons can, however, be made with previous Hamiltonian models. We provide a summary in Table 2. There are two main categories:**
>
> **SO(3) models (e.g. QHNet)** use direct tensor products, and scale poorly with the lmax (maximum spherical degree) dimension. Because of this, they cannot be scaled to a large number of atoms.
>
> **SO(2) convolution models (e.g. DeepH2, SLEM)** are more scalable with respect to lmax, but their examples are still relatively simple (< 100-200 atoms per unit cell) as the size of the graph used during training is limited by memory. So far, only our method has managed to train on and predict H for unit cells containing thousands of atoms due to efficient parallelization through strictly locality + augmented partitioning.
>
>
>  **Architecture** | **Approach** | **Max # Atoms per Unit Cell** |
> |-----------------|-------------|-----------------------------|
> | QHNet [6]      | SO(3)       | ~30                         |
> | WANet [7]      | **SO(2)**   | <200                        |
> | DeepH2 [3]     | **SO(2)**   | ≤150                        |
> | SLEM [8]       | **SO(2)**   | <100                        |
> | **This work**  | **SO(2)**   | 3000+                        |
>
> **Table 2:** Comparison of our model to state-of-the-art architectures on task, approach, and complexity of their respective training/testing datasets.
>
> **For performance comparison**, we achieved errors (0.99-5 meV) within the range of the values obtained by DeepH2 (2.2 meV) and HamGNN (1.5-3.23 meV),  despite our more challenging dataset. To further demonstrate what we can do with H predictions at such scale,**we provide a concrete application example in the response to Reviewer Kuqs.**
>
> **Graph partitioning literature**: The paper brought up by the reviewer (Mostafa) focuses on reducing the size of the computational graph during fully-distributed training. However, in our case the memory limits can already be overcome when processing the input graph embeddings, even without tracking gradients. The approaches are thus entirely different. To the best of our knowledge, no previous work on large graph partitioning(e.g., GraphSAGE, ClusterGCN [9]) has ever targeted quantitative predictions of materials properties, for which the graph connectivity must absolutely be preserved. Our partitioning approach is therefore also original in this regard.
>
> **Note that the list of references is found in the response to Reviewer 4 (nuFS)**

---

> > ### Comment · Reviewer_9oL5 · 2025-04-03
> >
> > Thank for the additional and clarifying details. Given this response, I am willing to increase my overall score to a 4. I would appreciate a deeper discussion on graph partitioning methods in the paper (can be in the appendix), which would encourage how works in adjacent fields relate to the present and potentially future work.

---

### Official Review · Reviewer_Kuqs · 2025-03-11

**Overall Recommendation:** 3

**Summary:**

The authors propose a method based on SO(2) graph neural networks to learn the electronic Hamiltonian matrix in the structural relaxation process of inorganic materials. To address computational challenges associated with defective or large-scale crystalline structures, they introduce an efficient partitioning strategy that enables the model to handle arbitrarily large structures while preserving local atomic environments.

**Claims And Evidence:**

The paper makes two primary contributions: (1) the development of a strictly local graph neural network (GNN)-based architecture for efficient prediction of the electronic Hamiltonian matrix; and (2) the introduction of an augmented partitioning method that enables large crystalline structures to be decomposed into smaller subgraphs while maintaining prediction accuracy comparable to global computation, thereby significantly reducing memory consumption.
For (1), the authors provide an extensive review of existing GNN-based Hamiltonian prediction methods in Section 2.2, but lacks a systematic comparison between the proposed architecture and previous methods, such as its advantages in accuracy, computational efficiency, and scalability.
For (2), I think the augmented partitioning method is well-supported with convincing evidence.

**Essential References Not Discussed:**

I think the authors have adequately cited and discussed prior work.

**Experimental Designs Or Analyses:**

The authors’ experimental design and analysis are comprehensive and align well with general methodologies in Hamiltonian matrix prediction research. The experiments involve multiple large-scale disordered crystalline datasets, and the partitioning-based prediction method is rigorously evaluated for both model performance and effectiveness.
However, the paper provides little discussion on the practical effectiveness of the method or its impact on computational efficiency. In the final sentence of Section 4.5, the authors mention that the proposed method could serve as an initial guess for DFT packages to reduce computational costs. Given that one of the key applications of Hamiltonian matrix prediction is to expedite electronic structure calculations, it would be beneficial for the authors to include a more detailed quantitative analysis.

**Methods And Evaluation Criteria:**

The authors construct three large-scale datasets of disordered crystalline materials and conduct a comprehensive analysis of prediction. Additionally, the authors employ three data augmentation strategies—atomic position perturbation, oxygen vacancy filling, and atomic substitution, to evaluate the model’s ability to capture both short-range and long-range interactions. The methodological design is robust.

**Other Comments Or Suggestions:**

No.

**Other Strengths And Weaknesses:**

Strengths: The proposed partitioning algorithm achieves good computational performance without sacrificing accuracy, particularly for large-scale disordered material systems.
Weaknesses: The paper lacks comparative analysis with other existing Hamiltonian prediction methods and does not provide quantitative analysis, such as how many SCF iterations can be reduced or how much speedup can be achieved in actual calculations.

**Questions For Authors:**

1. Application performance, see "Experimental Designs Or Analyses".
2. Model comparasion, see "Claims And Evidence".
3. How would this method perform when applying to small crystal structure (maybe a slice of the unit cell), is it possible to provide the prediction result analysis of a small slice of one structure from the proposed dataset (compared with previous works)?

**Relation To Broader Scientific Literature:**

In computational materials science, reducing computational time complexity while maintaining prediction accuracy has long been a key objective. In this work, the authors propose a partitioning-based method that addresses the challenge of directly handling large-scale disordered systems, offering promising applications in other disordered material systems.

**Theoretical Claims:**

I have reviewed the theoretical claims in the paper and have not identified any apparent issues.

---

> ### Author Rebuttal · Authors · 2025-03-28
>
> **1. Application performance**
>
> Using the predicted H as an initial guess for SCF iterations is not our main focus, and the reduction in iterations also depends on the convergence threshold. We instead focus on key downstream applications where the Hamiltonian matrix H is the final product needed. Examples include ab-initio quantum transport (QT) simulations of nanoscale devices where electrical current is computed or the investigation of the solubility of doping atoms in semiconductors with the nudged elastic band (NEB) method [1,2]. In these cases, obtaining the required H from DFT is a crucial step that can consume more than 90% of the computational time, particularly in structures made of 1000 atoms or more.
>
> **We provide here a concrete demonstration of our approach on a valence change memory (VCM) cell made of a TiN/HfO2/Ti/TiN stack and 5,268 atoms**. Normally, stoichiometric HfO2 is an insulator that blocks current flow. Applying a voltage introduces point defects (oxygen vacancies) that assemble into a conductive filament, changing the current by several orders of magnitude along the way.
>
> We first train a model on slices of two HfO2 structures with randomly distributed vacancies, and use it to predict the H of structures where vacancies were arranged into filaments. **As these filament shapes were not in the training data, this is an out-of-distribution generalization task.** The predicted H can then replace its DFT counterpart in ab-initio quantum transport simulations to compute the electrical current, as summarized in Table 1.
>
> | Device      | Node Error [mEₕ] | Edge Error [mEₕ] | Current (Label H) [A] | Current (Predicted H) [A] |
> |------------|------------|------------|----------------|----------------------|
> | Filament 0  | 1.81       | 0.20       | 1.98E-09      | 1.47E-09           |
> | Filament 1 | 1.74       | 0.20       | 1.12E-05      | 9.99E-06           |
> | Filament 2 | 1.66       | 0.20       | 1.45E-06      | 1.96E-06           |
>
> **Table 1:** Summary of prediction results and computed currents for devices with vacancy configurations forming different filament shapes.
>
> The currents computed with predicted H are well within acceptable error for downstream applications, given that the current values of the three filaments are on different orders of magnitude (1e-9, 1e-6 and 1e-5 A). They provide qualitative insight on the spatial localization of current flow, and allow us to study the reversible dielectric-breakdown process in HfO2 oxides. This has important implications, considering that direct ML inference is much faster than DFT (**~2 seconds for forward pass vs 3.94 node hours** on GH200 superchips).
>
> Firstly, we can enable previously unfeasible workflows involving repeated DFT updates. By replacing the DFT step with model inference, we can rapidly obtain the Hamiltonian matrices of different structures along, for example, an Molecular Dynamics (MD) trajectory, thus amortizing the training cost (<40 node hours) across 100s to 1000s of time steps. Besides filament formation in VCM, this can also be used to study electrical behavior during phase  transition of GST in memory devices. Secondly, we can perform inference on structures far larger than what DFT can handle (10k+ atoms). **It is therefore not only an efficient substitute for DFT, but also an essential step to unlocking new research areas.**
>
> **2. Model Comparison:**
>
> **Accuracy**: Firstly, slices cannot be isolated from a large structure to test other models due to periodic boundary conditions imposed during DFT. Interactions with periodic images cause the H elements of the isolated slices to be entirely different from those of the same atoms in bulk structures (MAE of 837 meV). The use of slices for training/testing is uniquely enabled by our strictly local model + augmented partitioning approach. Regardless, we can still compare our values with the accuracy achieved by SOTA architectures on smaller structures. Despite the complexity of our dataset comprising of disordered structures with thousands of atoms, we achieved errors of **0.99 to 5.16 meV** that are within the range of the values obtained by DeepH2 (**2.2 meV**) [3] and HamGNN (**1.5-3.23 meV**) [4]  for simpler periodic structures.
>
>
> **Scalability**: Note that many previous works use the term ‘scalable’ to refer to the l-max dimension, which can be tackled with SO(2) convolutions [3]. On top of this approach, our augmented partitioning method also enhances scalability with respect to the number of atoms/edges in the graph  (1k+ nodes ~ 100k+ edges), by allowing large graphs to be broken down and trained independently. To the best of our knowledge, **no other work has trained on graphs with 1k+ atoms**. We also provide analysis of the speedup offered by our approach when compared to conventional full graph training used by other models in **Appendix H**.
>
>
> **Note that the list of references is found in the response to Reviewer 3 (nuFS)**

---

### Decision · Program_Chairs · 2025-05-01

**Decision:**

Accept (poster)

**Comment:**

The paper presents a mean-field electronic Hamiltonian prediction model that can be applied to large systems. This arises from the need to process disordered materials, in which a huge unit cell should be assumed. The model implements this by assuming diagonal sparsity of the Hamiltonian matrix, restricting interatomic interaction to be within each segment, while keeping a few neighboring atoms ("virtual nodes") for each segment to account for inter-segment interaction. The authors applied the model on some real-world workloads with unit cells larger than before, and showed useful prediction results in terms of derived physical quantities.

All the reviewers and I myself appreciate the contribution of a useful Hamiltonian prediction model that is tractable on large atomic systems, based on a graph partitioning strategy and implementation. Reviewers also raised a few insufficiencies include seemingly minor technical novelty (using SO(2) convolution and assuming locality for scalability in Hamiltonian prediction are not new), and the lack of comparison with existing Hamiltonian prediction methods, even if only on a small-scale system.

Personally, I tend to value the contribution of connecting and adjusting prevailing machine learning techniques to a new (disordered material system) challenging (need of handling a large unit cell) practical need. The ablation studies of their own method provide a reasonable understanding on how various techniques and components work, which could also provide valuable information to the community. I won't judge too much on missing some evaluation settings/metrics as the selected ones seem more directly relevant to the real application, but I would encourage the authors to also provide such information to complete the analysis on the work (even if it shows a limitation).

As the rating of this submission appear a bit diverse, I posted the comment to the reviewers before decision. No other opinions were raised.